# Domain Agnostic Fourier Neural Operators

**Ning Liu**[*]
Global Engineering and
Materials, Inc.
Princeton, NJ 08540
ningliu@umich.edu

**Siavash Jafarzadeh**[*]
Department of Mathematics
Lehigh University
Bethlehem, PA 18015
sij222@lehigh.edu

**Yue Yu**[†]
Department of Mathematics
Lehigh University
Bethlehem, PA 18015
yuy214@lehigh.edu

## Abstract

Fourier neural operators (FNOs) can learn highly nonlinear mappings between
function spaces, and have recently become a popular tool for learning responses
of complex physical systems. However, to achieve good accuracy and efficiency,
FNOs rely on the Fast Fourier transform (FFT), which is restricted to modeling
problems on rectangular domains. To lift such a restriction and permit FFT on
irregular geometries as well as topology changes, we introduce domain agnostic
Fourier neural operator (DAFNO), a novel neural operator architecture for learning
surrogates with irregular geometries and evolving domains. The key idea is to
incorporate a smoothed characteristic function in the integral layer architecture of
FNOs, and leverage FFT to achieve rapid computations, in such a way that the
geometric information is explicitly encoded in the architecture. In our empirical
evaluation, DAFNO has achieved state-of-the-art accuracy as compared to baseline
neural operator models on two benchmark datasets of material modeling and airfoil
simulation. To further demonstrate the capability and generalizability of DAFNO
in handling complex domains with topology changes, we consider a brittle material
fracture evolution problem. With only one training crack simulation sample,
DAFNO has achieved generalizability to unseen loading scenarios and substantially
different crack patterns from the trained scenario. Our code and data accompanying
this paper are available at https://github.com/ningliu-iga/DAFNO.

## 1 Introduction

Deep learning surrogate models provide a useful data-driven paradigm to accelerate the PDE-solving
and calibration process of scientific modeling and computing problems. Among others, a wide range
of scientific computing applications entail the learning of solution operators, i.e., the learning of
infinite dimensional function mappings between any parametric dependence to the solution field.
A prototypical instance is the case of solving Navier-Stokes equations in fluid mechanics, where
the initial input needs to be mapped to a temporal sequence of nonlinear parametric solutions. The
demand for operator learning has sparked the development of neural operator based methods (Li
et al., 2020a,b,c; You et al., 2022a,b,c; Goswami et al., 2022; Liu et al., 2022; Gupta et al., 2021; Lu
et al., 2019; Cao, 2021; Hao et al., 2023; Li et al., 2022b; Yin et al., 2022c), with one of the most
popular architectures being Fourier Neural Operators (FNOs) (Li et al., 2020c; You et al., 2022c).

The success of FNOs can be mostly attributed to its convolution-based integral kernel that learns in a
resolution-invariant manner and the computationally efficient evaluation achieved via Fast Fourier
Transform (FFT) (Brigham, 1988). While learning in the spectral domain is fast, the latter comes at a
cost: the computational domain of the underlying problem needs to be rectangular with uniformly

---

[*]Equal contribution
[†]Corresponding author

37th Conference on Neural Information Processing Systems (NeurIPS 2023).

meshed grids. This is often intractable as the domain of interest is, more often than not, irregular. An often taken trick for applying FNO to irregular domains is to embed the original domain into a larger rectangular domain and zero-pad or extrapolate on the redundant space (Lu et al., 2022). This poses two potential problems, one being the possible numerical errors and even instabilities due to the discontinuity at the original domain boundary (e.g., the Gibbs phenomenon (Gottlieb & Shu, 1997)) and the other, perhaps more importantly, being the fact that the padding/extrapolating techniques cannot handle domains with shallow gaps, as is the case in object contact and crack propagation problems. Meanwhile, another line of work emphasizes the learning of a diffeomorphic mapping between the original domain and a latent domain with uniform grids on which FNO can be applied (Li et al., 2022a). However, in this approach the changes on the boundary and the domain topology can only be informed via the learned diffeomorphism, which results in approximation errors when tested on a new domain geometry and possible failure when a change in topology is involved on the domain geometry.

In this work, we aim to design FNO architectures that explicitly embed the boundary information of irregular domains, which we coin Domain Agnostic Fourier Neural Operator (DAFNO). This is inspired by the recent work in convolution-based peridynamics (Jafarzadeh et al., 2022b) in which bounded domains of arbitrary shapes are explicitly encoded in the nonlocal integral formulation. We argue that, by explicitly embedding the domain boundary information into the model architecture, DAFNO is able to learn the underlying physics more accurately, and the learnt model is generalizable to changes on the domain geometry and topology. Concretely, we construct two practical DAFNO variants, namely, eDAFNO that inherits the explicit FNO architecture (Li et al., 2022a) and iDAFNO that is built upon the implicit FNO (IFNO) architecture characterizing layer-independent kernels (You et al., 2022c). Moreover, a boundary smoothening technique is also proposed to resolve the Gibbs phenomenon and retain the fidelity of the domain boundary. In summary, the primary contributions of the current work are as follows:

- We propose DAFNO, a novel Fourier neural operator architecture that explicitly encodes the boundary information of irregular domains into the model architecture, so that the learned operator is aware of the domain boundary, and generalizable to different domains of complicated geometries and topologies.
- By incorporating a (smoothened) domain characteristic function into the integral layer, our formulation resembles a nonlocal model, such that the layer update acts as collecting interactions between material points inside the domain and cuts the non-physical influence outside the domain. As such, the model preserves the fidelity of the domain boundary as well as the convolution form of the kernel that retains the computational efficiency of FFT.
- We demonstrate the expressivity and generalizability of DAFNO across a wide range of scientific problems including constitutive modeling of hyperelastic materials, airfoil design, and crack propagation in brittle fracture, and show that our learned operator can handle not only irregular domains but also topology changes over the evolution of the solution.

## 2 Background and related work

The goal of this work is to construct a neural network architecture to learn common physical models on various domains. Formally, given $\mathcal{D} := \{(\boldsymbol{g}_i|_{\Omega_i}, \boldsymbol{u}_i|_{\Omega_i})\}_{i=1}^N$, a labelled set of function pair observations both defined on the domain $\boldsymbol{x} \in \Omega_i \subset \mathbb{R}^s$. We assume that the input $\{\boldsymbol{g}_i(\boldsymbol{x})\}$ is a set of independent and identically distributed (i.i.d.) random fields from a known probability distribution $\mu$ on $\mathcal{A}(\mathbb{R}^{d_g})$, a Banach space of functions taking values in $\mathbb{R}^{d_g}$. $\boldsymbol{u}_i(\boldsymbol{x}) \in \mathcal{U}(\mathbb{R}^{d_u})$, possibly noisy, is the observed corresponding response taking values in $\mathbb{R}^{d_u}$. Taking mechanical response modeling problem for example, $\Omega_i$ is the shape of the object of interest, $\boldsymbol{g}_i(\boldsymbol{x})$ may represent the boundary, initial, or loading conditions, and $\boldsymbol{u}_i(\boldsymbol{x})$ can be the resulting velocity, pressure, or displacement field of the object. We assume that all observations can be modeled by a common and possibly unknown governing law, e.g., balance laws, and our goal is to construct a surrogate operator mapping, $\tilde{\mathcal{G}}$, from $\mathcal{A}$ to $\mathcal{U}$ such that

$$\tilde{\mathcal{G}}[\boldsymbol{g}_i; \theta](\boldsymbol{x}) \approx \boldsymbol{u}_i(\boldsymbol{x}), \ \forall \boldsymbol{x} \in \Omega_i. \tag{1}$$

Here, $\theta$ represents the (trainable) network parameter set.

In real-world applications, the domain $\Omega_i$ can possess different topologies, e.g., in contact problems (Benson & Okazawa, 2004; Simo & Laursen, 1992) and material fragmentation problems (De Luycker et al., 2011; Agwai et al., 2011; Silling, 2003), and/or evolve with time, as is the case

in large-deformation problems (Shadden et al., 2010) and fluid–structure interaction applications (Kuhl et al., 2003; Kamensky et al., 2017). Hence, it is desired to develop an architecture with generalizability across various domains of complex shapes, so that the knowledge obtained from one geometry can be transferable to other geometries.

## 2.1 Learning solution operators of hidden physics

In real-world physical problems, predicting and monitoring complex system responses are ubiquitous in many applications. For these purposes, physics-based PDEs and tranditional numerical methods have been commonly employed. However, traditional numerical methods are solved for specific boundary, initial, and loading conditions $\boldsymbol{g}$ on a specific domain $\Omega$. Hence, the solutions are not generalizable to other domains and operating conditions.

To provide a more efficient and flexible surrogate model for physical response prediction, there has been significant progress in the development of deep neural networks (NNs) and scientific machine learning models (Ghaboussi et al., 1998, 1991; Carleo et al., 2019; Karniadakis et al., 2021; Zhang et al., 2018; Cai et al., 2022; Pfau et al., 2020; He et al., 2021; Besnard et al., 2006). Among others, neural operators (Li et al., 2020a,b,c; You et al., 2022a; Ong et al., 2022; Gupta et al., 2021; Lu et al., 2019, 2021b; Goswami et al., 2022; Tripura & Chakraborty, 2022) show particular promise in learning physics of complex systems: compared with classical NNs, neural operators are resolution independent and generalizable to different input instances. Therefore, once the neural operator is trained, solving for a new instance of the boundary/initial/loading condition with a different discretization only requires a forward pass. These advantages make neural operators a useful tool to many physics modeling problems (Yin et al., 2022a; Goswami et al., 2022; Yin et al., 2022b; Li et al., 2020a,b,c; Lu et al., 2022, 2021a).

## 2.2 Neural operator learning

Here, we first introduce the basic architecture of the general integral neural operators (Li et al., 2020a,b,c; You et al., 2022a,c), which are comprised of three building blocks. First, the input function, $\boldsymbol{g}(\boldsymbol{x}) \in \mathcal{A}$, is lifted to a higher-dimensional representation via $\boldsymbol{h}^0(\boldsymbol{x}) = \mathcal{P}[\boldsymbol{g}](\boldsymbol{x}) := P[\boldsymbol{x}, \boldsymbol{g}(\boldsymbol{x})]^T + \boldsymbol{p}$, where $P \in \mathbb{R}^{(s+d_g) \times d_h}$ and $\boldsymbol{p} \in \mathbb{R}^{d_h}$ define an affine pointwise mapping. Then, the feature vector function $\boldsymbol{h}^0(\boldsymbol{x})$ goes through an iterative layer block, where the layer update is defined via the sum of a local linear operator, a nonlocal integral kernel operator, and a bias function: $\boldsymbol{h}^{l+1}(\boldsymbol{x}) = \mathcal{J}^{l+1}[\boldsymbol{h}^l](\boldsymbol{x})$. Here, $\boldsymbol{h}^l(\boldsymbol{x}) \in \mathbb{R}^{d_h}$, $l = 0, \cdots, L$, is a sequence of functions representing the values of the network at each hidden layer. $\mathcal{J}^1, \cdots, \mathcal{J}^L$ are the nonlinear operator layers defined by the particular choice of networks. Finally, the output $\boldsymbol{u}(\boldsymbol{x}) \in \mathcal{U}$ is obtained via a projection layer by mapping the last hidden layer representation $\boldsymbol{h}^L(\boldsymbol{x})$ onto $\mathcal{U}$ as: $\boldsymbol{u}(\boldsymbol{x}) = \mathcal{Q}[\boldsymbol{h}^L](\boldsymbol{x}) := Q_2 \sigma(Q_1 \boldsymbol{h}^L(\boldsymbol{x}) + \mathbf{q}_1) + \mathbf{q}_2$. $Q_1, Q_2, \mathbf{q}_1$ and $\mathbf{q}_2$ are the appropriately sized matrices and vectors that are part of the learnable parameter set, and $\sigma$ is an activation function (e.g., ReLU (He et al., 2018) or GeLU).

Then, the system response can be learnt by constructing a surrogate operator of (1): $\tilde{\mathcal{G}}[\boldsymbol{g}; \theta](\boldsymbol{x}) := \mathcal{Q} \circ \mathcal{J}^1 \circ \cdots \circ \mathcal{J}^L \circ \mathcal{P}[\boldsymbol{g}](\boldsymbol{x}) \approx \boldsymbol{u}(\boldsymbol{x})$, by solving the network parameter set $\theta$ via an optimization problem:

$$\min_{\theta \in \Theta} \mathcal{L}_{\mathcal{D}}(\theta) := \min_{\theta \in \Theta} \sum_{i=1}^{N} [C(\tilde{\mathcal{G}}[\boldsymbol{g}_i; \theta], \boldsymbol{u}_i)] . \tag{2}$$

Here, $C$ denotes a properly defined cost functional (e.g., the relative mean square error) on $\Omega_i$.

## 2.3 Fourier neural operators

The Fourier neural operator (FNO) is first proposed in Li et al. (2020c) with its iterative layer architecture given by a convolution operator:

$$\mathcal{J}^l[\boldsymbol{h}](\boldsymbol{x}) := \sigma \left( W^l \boldsymbol{h}(\boldsymbol{x}) + \boldsymbol{c}^l + \int_{\Omega} \kappa(\boldsymbol{x} - \boldsymbol{y}; \boldsymbol{v}^l) \boldsymbol{h}(\boldsymbol{y}) d\boldsymbol{y} \right) , \tag{3}$$

where $W^l \in \mathbb{R}^{d_h \times d_h}$ and $\boldsymbol{c}^l \in \mathbb{R}^{d_h}$ are learnable tensors at the $l$-th layer, and $\kappa \in \mathbb{R}^{d_h \times d_h}$ is a tensor kernel function with parameters $\boldsymbol{v}^l$. When a rectangular domain $\Omega$ with uniform meshes is considered, the above convolution operation can be converted to a multiplication operation through discrete Fourier transform:

$$\mathcal{J}^l[\boldsymbol{h}](\boldsymbol{x}) = \sigma \left( W^l \boldsymbol{h}(\boldsymbol{x}) + \boldsymbol{c}^l + \mathcal{F}^{-1}[\mathcal{F}[\kappa(\cdot; \boldsymbol{v}^l)] \cdot \mathcal{F}[\boldsymbol{h}(\cdot)]](\boldsymbol{x}) \right) ,$$

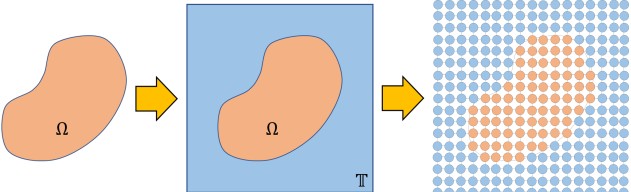

Figure 1: A schematic of extending an arbitrarily shaped domain $\Omega$ to a periodic box $\mathbb{T}$ and its discretized form in 2D.

where $\mathcal{F}$ and $\mathcal{F}^{-1}$ denote the Fourier transform and its inverse, respectively, which are computed using the FFT algorithm to each component of $\boldsymbol{h}$ separately. The FFT calculations greatly improve the computational efficiency due to their quasilinear time complexity, but they also restrict the vanilla FNO architecture to rectangular domains $\Omega$ (Lu et al., 2022).

To enhance the flexibility of FNO in modeling complex geometries, in Lu et al. (2022) the authors proposed to pad and/or extrapolate the input and output functions into a larger rectangular domain. However, such padding/extrapolating techniques are prone to numerical instabilities (Gottlieb & Shu, 1997), especially when the domain is concave and/or with complicated boundaries. As shown in Lu et al. (2022), the performance of dgFNO+ substantially deteriorates when handling complicated domains with notches and gaps. In Geo-FNO (Li et al., 2022a), an additional neural network is employed and trained from data, to continuously map irregular domains onto a latent space of rectangular domains. As a result, the vanilla FNO can be employed on the rectangular latent domain. However, this strategy relies on the continuous mapping from the physical domain to a rectangular domain, hence it is restricted to relatively simple geometries with no topology change.

## 3 Domain Agnostic Fourier Neural Operators

In this section, we introduce Domain Agnostic Fourier Neural Operator (DAFNO), which features the generalizability to new and unseen domains of arbitrary shapes and different topologies. The key idea is to explicitly encode the domain information in the design while retaining the convolutional architecture in the iterative layer of FNOs. In what follows, we present the eDAFNO architecture based on the standard/explicit FNO model (Li et al., 2020c), while the iDAFNO architecture based on the implicit FNO model (You et al., 2022c) is provided in Appendix A.

Concretely, we enclose the physical domain of interest, $\Omega$, by a (slightly larger) periodic box $\mathbb{T}$, as shown in Figure 1. Next, we define the following domain characteristic function:

$$\chi(\boldsymbol{x}) = \begin{cases} 1 & \boldsymbol{x} \in \Omega \\ 0 & \boldsymbol{x} \in \mathbb{T} \setminus \Omega \end{cases}, \tag{4}$$

which encodes the domain information of different geometries. Inspired by Jafarzadeh et al. (2022b), we incorporate the above-encoded domain information into the FNO architecture of (3), by multiplying the integrand in its convolution integral with $\chi(\boldsymbol{x})\chi(\boldsymbol{y})$:

$$\mathcal{J}^l[\boldsymbol{h}] = \sigma\left(\int_{\mathbb{T}} \chi(\boldsymbol{x})\chi(\boldsymbol{y})\kappa(\boldsymbol{x} - \boldsymbol{y}; \boldsymbol{v}^l)(\boldsymbol{h}(\boldsymbol{y}) - \boldsymbol{h}(\boldsymbol{x}))d\boldsymbol{y} + W^l\boldsymbol{h}(\boldsymbol{x}) + \boldsymbol{c}^l\right). \tag{5}$$

Herein, we have followed the practice in You et al. (2022a) and reformulated (3) to a nonlocal Laplacian operator, which is found to improve training efficacy. By introducing the term $\chi(\boldsymbol{x})\chi(\boldsymbol{y})$, the integrand vanishes when either point $\boldsymbol{x}$ or $\boldsymbol{y}$ is positioned inside $\Omega$ and the other is positioned outside. This modification eliminates any undesired interaction between the regions inside and outside of $\Omega$. As a result, it tailors the integral operator to act on $\Omega$ independently and is able to handle different domains and topologies. With this modification, the FFT remains applicable, since the convolutional structure of the integral is preserved and the domain of operation yet spans to the whole rectangular box $\mathbb{T}$. In this context, (5) can be re-organized as:

$$\mathcal{J}^l[\boldsymbol{h}] = \sigma\left(\chi(\boldsymbol{x})\left(\int_{\mathbb{T}} \kappa(\boldsymbol{x} - \boldsymbol{y}; \boldsymbol{v}^l)\chi(\boldsymbol{y})\boldsymbol{h}(\boldsymbol{y})d\boldsymbol{y} - \boldsymbol{h}(\boldsymbol{x})\int_{\mathbb{T}} \kappa(\boldsymbol{x} - \boldsymbol{y}; \boldsymbol{v}^l)\chi(\boldsymbol{y})d\boldsymbol{y} + W^l\boldsymbol{h}(\boldsymbol{x}) + \boldsymbol{c}^l\right)\right).$$

Note that multiplying $W^l\boldsymbol{h}(\boldsymbol{x}) + \boldsymbol{c}^l$ with $\chi(\boldsymbol{x})$ does not alter the computational domain. Now that the integration region is a rectangular box $\mathbb{T}$, the FFT algorithm and its inverse can be readily applied,

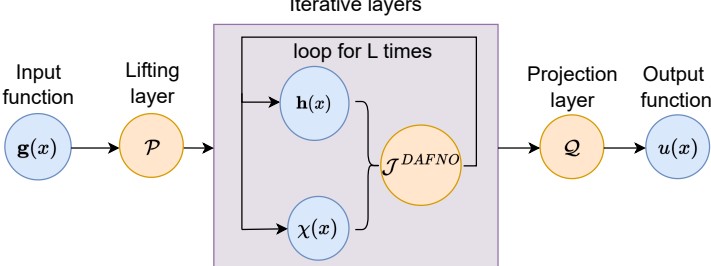

Figure 2: An illustration of the proposed DAFNO architecture. We start from the input function $\boldsymbol{g}(x)$. After lifting, the iterative Fourier layers are built that explicitly embed the encoded domain information, $\chi$. Lastly, we project the last hidden layer representation to the target function space.

and hence we can further express the eDAFNO architecture as:

$$\mathcal{J}^l[\boldsymbol{h}] := \sigma\left(\chi(\boldsymbol{x})\big(\mathcal{I}(\chi(\cdot)\boldsymbol{h}(\cdot); \boldsymbol{v}^l) - \boldsymbol{h}(\boldsymbol{x})\mathcal{I}(\chi(\cdot); \boldsymbol{v}^l) + W^l\boldsymbol{h}(\boldsymbol{x}) + \boldsymbol{c}^l\big)\right),$$

$$\text{where} \quad \mathcal{I}(\circ; \boldsymbol{v}^l) := \mathcal{F}^{-1}\big[\mathcal{F}[\kappa(\cdot; \boldsymbol{v}^l)] \cdot \mathcal{F}[\circ]\big] .$$

(6)

An illustration of the DAFNO atchiecture is provided in Figure 2. Note that this architectural modification is performed at the continuum level and therefore is independent of the discretization. Then, the box $\mathbb{T}$ can be discretized with structured grids (cf. Figure 1), as is the case in standard FNO.

Although the proposed DAFNO architecture in (6) can handle complex generalization in domains, it has a potential pitfall: since the characteristic function is not continuous on the domain boundary, its Fourier series cannot converge uniformly and the FFT result would present fictitious wiggling near the discontinuities (i.e., the Gibbs phenomenon (Day et al., 1965)). As a consequence, the introduction of $\chi$ can potentially jeopardize the computational accuracy. To improve the efficacy of DAFNO, we propose to replace the sharp characteristic function, $\chi$, with a smoothed formulation:

$$\tilde{\chi}(\boldsymbol{x}) := \tanh(\beta\text{dist}(\boldsymbol{x}, \partial\Omega))(\chi(\boldsymbol{x}) - 0.5) + 0.5 .$$

(7)

Here, the hyperbolic tangent function $\tanh(z) := \frac{\exp(z) - \exp(-z)}{\exp(z) + \exp(-z)}$, $\text{dist}(\boldsymbol{x}, \partial\Omega)$ denotes the (approximated) distance between $\boldsymbol{x}$ and the boundary of domain $\Omega$, and $\beta$ controls the level of smoothness, which is treated as a tunable hyperparameter. An illustration of the effect of the smoothed $\tilde{\chi}$ is displayed in Figure 3, with additional plots and prediction results with respect to different levels of smoothness, $\beta$, provided in Appendix B.1. In what follows, the $\tilde{\cdot}$ sign is neglected for brevity.

**Remark:** We point out that the proposed smoothed geometry encoding technique, although simple, is substantially different from existing function padding/extrapolation techniques proposed in Lu et al. (2022) who cannot handle singular boundaries (as in the airfoil tail of our example 2) nor notches/shallow gaps in the domain (as in the crack propagation of our example 3). Our proposed architecture in (5) is also more sophiscated than a trivial zero padding at each layer in that the characteristic function $\chi(\boldsymbol{x})$ is multiplied with the integrand, whereas the latter breaks the convolutional structure and hinders the application of FFTs.

## 4 Numerical examples

In this section, we demonstrate the accuracy and expressivity of DAFNO across a wide variety of scientific problems. We compare the performance of DAFNO against other relevant scientific machine learning models, including FNO (Li et al., 2020c), Geo-FNO (Li et al., 2022a), IFNO (You et al., 2022c), F-FNO (Tran et al., 2022), GNO (Li et al., 2020a), DeepONet (Lu et al., 2019), and UNet (Ronneberger et al., 2015). In particular, we carry out three experiments on irregular domains, namely, constitutive modeling of hyperelasticity in material science, airfoil design in fluid mechanics, and crack propagation with topology change in fracture mechanics. For fair comparison, the hyperparameters of each model are tuned to minimize the error on validation datasets, including initial learning rate, decay rate for every 100 epochs, smoothing parameter, and regularization parameter, while the total number of epochs is restricted to 500 for computational efficiency. The

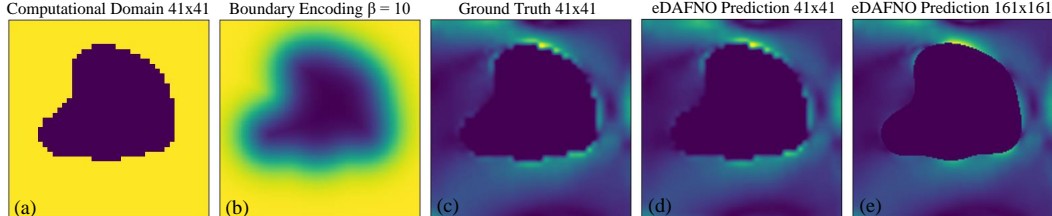

| Computational Domain 41x41 | Boundary Encoding β = 10 | Ground Truth 41x41 | eDAFNO Prediction 41x41 | eDAFNO Prediction 161x161 |
| (a) | (b) | (c) | (d) | (e) |

Figure 3: An illustration on a hyperelasticity sample: (a) sharp characteristic function $\chi$, (b) smoothed characteristic function $\chi$, (c) ground truth, (d) eDAFNO (trained using 41×41 discretization) prediction from the same resolution, and (e) zero-shot super-resolution prediction from eDAFNO (trained using 41×41 discretization and evaluated directly on 161×161 discretization).

relative L2 error is reported as comparison metrics for both the training and test datasets. The experiments of each method are repeated on 5 trials with 5 different random seeds, and the mean and standard deviation of the errors are reported. Further details and additional results are provided in Appendix B.

## 4.1 Constitutive modeling of hyperelasticity

We start with a hyperelasticity problem in material science that models the fundamental principle of constitutive relations. The high-fidelity synthetic data in this benchmark is governed by

$$\rho \frac{\partial^2 \boldsymbol{u}}{\partial t^2} + \nabla \cdot \boldsymbol{\sigma} = 0 \,, \tag{8}$$

where $\rho$ denotes the mass density, $\boldsymbol{u}$ and $\boldsymbol{\sigma}$ represent the corresponding displacement and stress fields, respectively. The computational domain is enclosed by a unit cell $[0,1]^2$ of which the center exists a randomly shaped void, as described by its radius $r = 0.2 + \frac{0.2}{1+exp(\tilde{r})}$ and $\tilde{r} \sim \mathbb{N}(0, 4^2(-\nabla + 3^2)^{-1})$. The bottom edge is fixed and the top edge is subjected to a tensile traction of $\boldsymbol{t} = [0, 100]$. The underlying hyperelastic material is of the incompressible Rivlin-Saunders type. For training, we directly adopt the dataset in Li et al. (2022a), where a total of 1000, 200, 200 samples are selected for training, validation and testing, respectively. For this problem, the input is represented as point clouds and the target is the resulting stress field.

Table 1: The total number of parameters (in millions) of selected models for hyperelasticity dataset.

| model | eDAFNO | iDAFNO | FNO | IFNO | Geo-FNO | GNO | DeepONet | UNet | F-FNO |
|---|---|---|---|---|---|---|---|---|---|
| nparam | 2.37 | 0.60 | 2.37 | 0.60 | 3.02 | 2.64 | 3.10 | 3.03 | 3.21 |

Table 2: Test errors for the hyperelasticity problem, where bold numbers highlight the best method.

| Model | | # of training samples | | |
|---|---|---|---|---|
| | | 10 | 100 | 1000 |
| Proposed model | eDAFNO | **16.446%±0.472%** | 4.247%±0.066% | **1.094%±0.012%** |
| | iDAFNO | 16.669%±0.523% | **4.214%±0.058%** | 1.207%±0.006% |
| Baseline model | FNO w/ mask | 19.487%±0.633% | 7.852%±0.130% | 4.550%±0.062% |
| | IFNO w/ mask | 19.262%±0.376% | 7.700%±0.062% | 4.481%±0.022% |
| | Geo-FNO | 28.725%±2.600% | 10.343%±4.446% | 2.316%±0.283% |
| | GNO | 29.305%±0.321% | 18.574%±0.584% | 13.007%±0.729% |
| | DeepONet | 35.334%±0.179% | 25.455%±0.245% | 11.998%±0.786% |
| | F-FNO | 35.672%±3.852% | 12.135%±5.813% | 3.193%±1.622% |
| | UNet | 98.167%±0.236% | 34.467%±2.858% | 5.462%±0.048% |
| Ablation study | FNO w/ smooth $\chi$ | 17.431%±0.536% | 5.479%±0.186% | 1.415%±0.025% |
| | IFNO w/ smooth $\chi$ | 17.145%±0.432% | 5.088%±0.146% | 1.509%±0.018% |

**Ablation study** We first carry out an ablation study by comparing (a) the proposed two DAFNO models with (b) the baseline FNO/IFNO with the sharp characteristic function as input (denoted as FNO/IFNO w/ mask), and (c) the original FNO/IFNO with our proposed smoothened boundary

characteristic function as input (denoted as FNO/IFNO w/ smooth $\chi$). In this study, scenarios (b) and (c) aim to investigate the effects of our proposed boundary smoothing technique, and by comparing scenarios (a) and (c) we verify the effectiveness of encoding the boundary information in eDAFNO architecture. In addition, both FNO- and IFNO-based models are tested, with the purpose to evaluate the model expressivity when using layer-independent parameters in iterative layers. Three training dataset sizes (i.e., 10, 100, and 1000) are employed to explore the effect of the proposed algorithms on small, medium, and large datasets, respectively. The number of trainable parameters are reported in Table 1. Following the common practice as in Li et al. (2022a), the hyperparameter choice of each model is selected by tuning the number of layers and the width (channel dimension) keeping the total number of parameters of the same magnitude.

The results of the ablation study are listed in Table 2. Firstly, by directly comparing the results of FNO (and IFNO) with mask and with smooth boundary encoding, one can tell that the boundary smoothing technique helps to reduce the error. This is supported by the observation that FNO and IFNO with smooth $\chi$ consistently outperform their counterparts with mask in all data regimes, especially when a sufficient amount of data becomes available where a huge boost in accuracy can be achieved (by over 300%). On the other hand, by encoding the geometry information into the iterative layer, the prediction accuracy is further improved, where eDAFNO and iDAFNO outperform FNO and IFNO with smoothed $\chi$ by 22.7% and 20.0% in large-data regime, respectively. Another interesting finding is that eDAFNO is 9.4% more accurate compared to iDAFNO in the large-data regime, although only a quarter of the total number of parameters is needed in iDAFNO due to its layer-independent parameter setting. This effect is less pronounced as we reduce the amount of available data for training, where the performance of iDAFNO is similar to that of eDAFNO in the small- and medium-data regimes. This is because iDAFNO has a much smaller number of trainable parameters and therefore is less likely to overfit with small datasets. Given that the performance of eDAFNO and iDAFNO is comparable, it is our opinion that both architectures are useful in different applications. In Figure 3, an example of the computational domain, the smoothened boundary encoding, the ground truth solution, and the eDAFNO prediction are demonstrated. To demonstrate the capability of prediction across resolutions, we train eDAFNO using data with $41\times41$ grids then apply the model to provide prediction on $161\times161$ grids–one can see that eDAFNO can generalize across different resolutions.

**Comparison against additional baselines** We further compare the performance of DAFNO against additional relevant baselines, including GNO, Geo-FNO, F-FNO, DeepONet, and UNet. Note that the results of GNO, DeepONet, and UNet are obtained using the same settings as in Li et al. (2022a). Overall, the two DAFNO variants are significantly superior to other baselines in accuracy, with eDAFNO outperforming GNO, DeepONet, UNet, Geo-FNO, and F-FNO by 1088.9%, 975.0%, 399.3%, 111.7%, and 191.9%, respectively, in large-data regime. DAFNO is also more memory efficient compared to Geo-FNO (the most accurate baseline), as it foregoes the need for additional coordinate deformation network. As shown in Table 1, when the layer-independent parameter setting in iDAFNO is taken into account, DAFNO surpasses Geo-FNO by 407.4% in memory saving.

## 4.2 Airfoil design

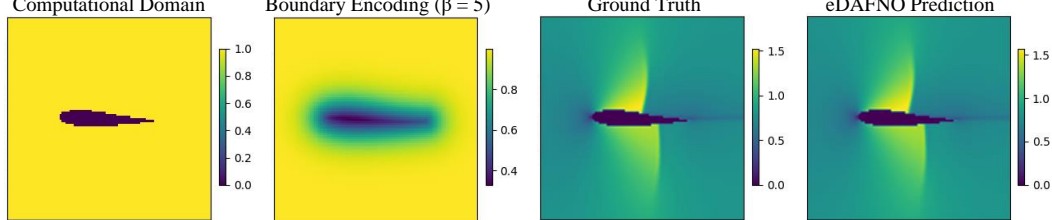

Figure 4: An illustration on a test sample from the airfoil design problem. From left to right: an illustration of the discretized computational domain, the smoothed boundary encoding (i.e., smoothed $\chi$), the ground truth, and the eDAFNO prediction.

In this example, we investigate DAFNO's performance in learning transonic flow around various airfoil designs. Neglecting the effect of viscosity, the underlying physics can be described by the Euler equation:

$$\frac{\partial \rho}{\partial t} + \nabla \cdot (\rho \boldsymbol{v}) = 0 \,, \quad \frac{\partial \rho \boldsymbol{v}}{\partial t} + \nabla \cdot (\rho \boldsymbol{v} \otimes \boldsymbol{v} + p\mathbb{I}) = 0 \,, \quad \frac{\partial E}{\partial t} + \nabla \cdot ((E+p)\,\boldsymbol{v}) = 0 \,, \quad (9)$$

Table 3: Results for the airfoil design problem, where bold numbers highlight the best method.

| | Model | Train error | Test error |
|---|---|---|---|
| Proposed model | eDAFNO | 0.329%±0.020% | **0.596%±0.005%** |
| | iDAFNO | 0.448%±0.012% | 0.642%±0.020% |
| | eDAFNO on irregular grids | 0.331%±0.003% | 0.659%±0.007% |
| Baseline model | Geo-FNO | 1.565%±0.180% | 1.650%±0.175% |
| | F-FNO | 0.566%±0.066% | 0.794%±0.025% |
| | FNO w/ mask | 2.676%±0.054% | 3.725%±0.108% |
| | UNet w/ mask | 2.781%±1.084% | 4.957%±0.059% |

with $\rho$, $p$ and $E$ being the fluid density, pressure and the total energy, respectively, and $v$ denoting the corresponding velocity field. The applied boundary conditions are: $\rho_\infty = 1$, Mach number $M_\infty = 0.8$, and $p_\infty = 1$ on the far field, with no penetration enforced on the airfoil. The dataset used for training is directly taken from Li et al. (2022a), which consists of variations of the NACA-0012 airfoil and is divided into 1000, 100, 100 samples for training, validation and testing, respectively. For this problem, we aim to learn the resulting Mach number field based on a given mesh as input. An example of the computational domain, the smoothened boundary encoding, the ground truth, and the eDAFNO prediction is illustrated in Figure 4.

We report in Table 3 our experimental observations using eDAFNO and iDAFNO, along with the comparison against FNO, Geo-FNO, F-FNO and UNet, whose models are directly attained from Li et al. (2022a); Tran et al. (2022). We can see that the proposed eDAFNO achieves the lowest error on the test dataset, beating the best result of non-DAFNO baselines by 24.9% and Geo-FNO by 63.9%, respectively. Additionally, iDAFNO reaches a similar level of accuracy with only a quarter of the total number of parameters employed, which is consistent with the findings in the previous example. With the properly trained DAFNO models, efficient optimization and inverse design are made possible.

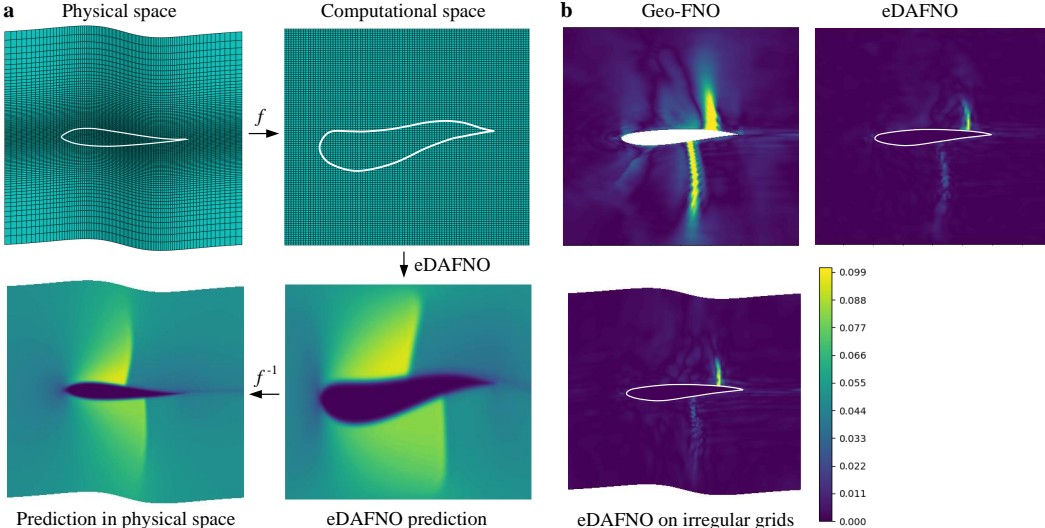

Figure 5: eDAFNO applied to the airfoil dataset on irregular grids. (a): the highly irregular and adaptive grids in the physical space is firstly deformed (via either an analytical mapping $f$ or a trainable neural network for grid deformation) to uniform grids in the computational space, on which eDAFNO is applied to learn the underlying physics. The learned prediction is then converted back to the physical space via the inverse mapping $f^{-1}$ or with another trainable neural network. (b): an illustration of the absolute error distribution of predictions in Geo-FNO, eDAFNO trained on uniform grids, and eDAFNO trained on irregular grids.

Aside from the enhanced predictability, DAFNO can also be easily combined with the grid mapping technique in Geo-FNO to handle non-uniform grids. In particular, no modification on the model architecture is required, where one just needs to include an analytical or trainable mapping from

non-uniform/irregular mesh grids to uniform mesh grids. As a demonstration, we consider irregular grids of the airfoil dataset and use a pre-computed function to map irregular grids to regular grids. In the irregular grid set, we place more grid points near the airfoil to provide a better resolution near the important parts. The test error is provided in Table 3, where the test loss using the eDAFNO learned model is $0.659\% \pm 0.007\%$, which is similar to the DAFNO results on uniform grids. In Figure 5, we demonstrate the irregular mesh and the absolute error comparison across Geo-FNO, DAFNO on regular grids, and DAFNO on irregular grids. One can see that, while both DAFNOs substantially outperform Geo-FNO, the error contours from eDAFNO with an irregular mesh show a smaller miss-match region near the airfoil, illustrating the flexibility of DAFNO in meshing and its capability in resolving fine-grained features.

### 4.3  Crack propagation with topology change in domain

In this example, we aim to showcase DAFNO's capability in handling evolving domains by modeling crack propagation in brittle fracture. We emphasize that DAFNO represents the first neural operator that allows for learning with topology change. In the field of brittle fracture, a growing crack can be viewed as a change in topology, which corresponds to an evolving $\chi(t)$ in the DAFNO architecture. In particular, we define the following time-dependent characteristic function:

$$\chi(\boldsymbol{x}, t) = \begin{cases} 1 & \boldsymbol{x} \in \Omega(t) \\ 0 & \boldsymbol{x} \in \mathbb{T} \setminus \Omega(t) \end{cases}, \tag{10}$$

where $\Omega(t)$ denotes the time-dependent domain/evolving topology. Employing time-dependent $\chi$ in (6) keeps the neural operator informed about the evolving topology. In general, the topology evolution rule that determines $\Omega(t)$ can be obtained from a separate neural network or from physics as is the case in the current example. The high-fidelity synthetic data in this example is generated using 2D PeriFast software (Jafarzadeh et al., 2022a), which employs a peridynamics (PD) theory for modeling fracture (Bobaru et al., 2016). A demonstration of the evolving topology, as well as further details regarding the governing equations and data generation strategies, is provided in Appendix B.3.

In this context, we select eDAFNO as the surrogate model to learn the the internal force density operator. Specifically, let $u_1(\boldsymbol{x}, t)$, $u_2(\boldsymbol{x}, t)$ denote the two components of the displacement field $\boldsymbol{u}$ at time $t$, and $L_1$, $L_2$ be the two components of the internal force density $\mathcal{L}[\boldsymbol{u}]$. Given $u_1$, $u_2$ and $\chi$ as input data, we train two separate eDAFNO models to predict $L_1$ and $L_2$, respectively. Then, we substitute the trained surrogate models in the dynamic governing equation and adopt Velocity–Verlet time integration to update $\boldsymbol{u}$ for the next time step, whereafter $\Omega$ and $\chi$ are updated accordingly.

The problem of interest is defined on a 40 mm $\times$ 40 mm thin plate with a pre-crack of length 10 mm at one edge, which is subjected to sudden, uniform, and constant tractions on the top and bottom edges. Depending on the traction magnitude, crack grows at different speeds, may or may not bifurcate, and the final crack patterns can be of various shapes. Our training data is comprised of two parts, one consisting of 450 snapshots from a crack propagation simulation with a fixed traction magnitude of $\sigma = 4$ MPa, and the other consisting of randomized sinusoidal displacement fields and the corresponding $L_1$, $L_2$ fields computed by the PD operator. The sinusoidal subset contains 4,096 instances without fracture and is used to diversify the training data and mitigate overfitting on the crack data. For testing, we use the trained models in two scenarios with traction magnitudes different from what is used in training, which allows us to evaluate the generalizability of eDAFNO. Note that these tests are highly challenging for the trained models, as the predictions of the previous time step are used as the models' input in the current step, which leads to error accumulation as the simulation marches forward.

Figure 6 displays the test results on the crack patterns under different traction magnitudes, where the low traction magnitude (left column) shows a slow straight crack, the mid-level traction results in a crack with a moderate speed and a bifurcation event (middle column), and the highest traction leads to a rapid crack growth and initiates a second crack from the other side of the plate (right column). The trained eDAFNO model is able to accurately predict the left and right examples while only seeing the middle one, indicating that it has correctly learned the true constitutive behavior of the material and generalized well to previously unseen loading scenarios and the correspondingly changed domain topology. To further verify eDAFNO's generalizability to unseen geometries, in Figure 7 we compare eDAFNO with the baseline FNO w/ smoothed mask model, and plot their relative errors in $\chi$ and $\boldsymbol{u}$. As observed, the FNO predictions become unstable at a fairly early time in the two test scenarios,

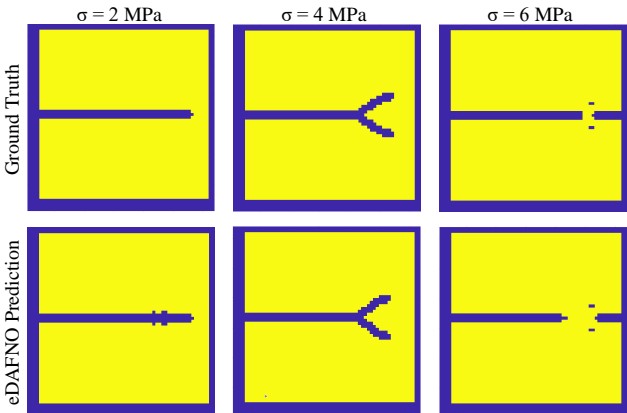

Figure 6: Comparison of the fracture patterns with different loading scenarios between the high-fidelity solution and eDAFNO prediction.

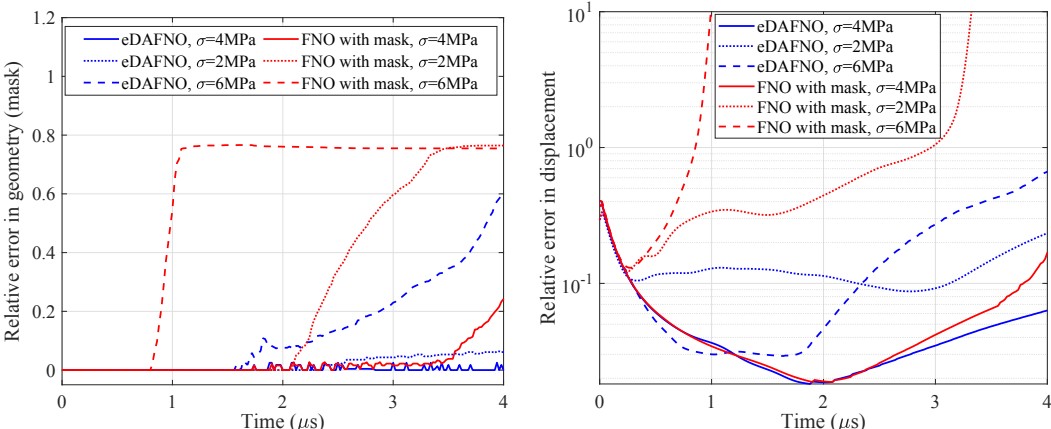

Figure 7: Comparison of relative errors in the characteristic function $\chi$ (left) and displacement fields (right) of the evolving topology predicted using the trained eDAFNO and FNO with mask at each time step for the training (loading = 4 MPa) and testing (loading = 2 MPa and 6 MPa) scenarios.

since it creates a mask which is out of training distribution, while DAFNO can handle different crack patterns by hard coding it in the architecture, so the results remain stable for a much longer time.

## 5  Conclusion

We introduce two DAFNO variants to enable FNOs on irregular domains for PDE solution operator learning. By incorporating the geometric information from a smoothed characteristic function in the iterative Fourier layer while retaining the convolutional form, DAFNO possesses the computational efficiency from FFT together with the flexibility to operate on different computational domains. As a result, DAFNO is not only highly efficient and flexible in solving problems involving arbitrary shapes, but it also manifests its generalizability on evolving domains with topology change. In two benchmark datasets and a real-world crack propagation dataset, we demonstrate the state-of-the-art performance of DAFNO. We find both architectures helpful in practice: eDAFNO is slightly more accurate while iDAFNO is more computationally efficient and less overfitting with limited data.

**Limitation:** Due to the requirement of the standard FFT package, in the current DAFNO we focus on changing domains with uniformly meshed grids. However, we point out that this limitation can be lifted by using nonuniform FFT (Greengard & Lee, 2004) or an additional mapping for grid deformation, as shown in the airfoil experiment. Additionally, in our applications, we have focused on applying the same types of boundary conditions to the changing domains (e.g., all Dirichlet or all Neumann). In this context, another limitation and possible future direction would be on the transferability to PDEs with different types of boundary conditions.

## Acknowledgments and Disclosure of Funding

S. Jafarzadeh would like to acknowledge support by the AFOSR grant FA9550-22-1-0197, and Y. Yu would like to acknowledge support by the National Science Foundation under award DMS-1753031 and the AFOSR grant FA9550-22-1-0197. Portions of this research were conducted on Lehigh University's Research Computing infrastructure partially supported by NSF Award 2019035.

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
