# A  Detailed iDAFNO architecture

Similar to the eDAFNO architecture shown in (6), we present the iDAFNO version by incorporating the layer-independent parameter definition characterized in the IFNO structure (You et al., 2022c):

$$\mathcal{J}[\boldsymbol{h}](\boldsymbol{x}) := \boldsymbol{h}(\boldsymbol{x}) + \tau\sigma\bigg(\chi(\boldsymbol{x})\big(\mathcal{I}(\chi(\cdot)\boldsymbol{h}(\cdot);\boldsymbol{v}) - \boldsymbol{h}(\boldsymbol{x})\mathcal{I}(\chi(\cdot);\boldsymbol{v}) + W\boldsymbol{h}(\boldsymbol{x}) + \boldsymbol{c}\big)\bigg),$$

$$\text{where}\quad \mathcal{I}(\circ;\boldsymbol{v}) := \mathcal{F}^{-1}\big[\mathcal{F}[\kappa(\cdot;\boldsymbol{v})]\cdot\mathcal{F}[\circ]\big].$$
(11)

Here, $\tau = \dfrac{1}{L}$ is the reciprocal of the total number of layers employed. Note that the superscript $l$ is dropped because the model parameters are layer-independent in the iDAFNO architecture, which leads to significant computational saving.

# B  Problem settings and additional experimental results

In order to maintain consistency with other baselines, the dimension of representation in the first two examples is set to $d_h = 32$, with a total of 4 Fourier layers and 12 Fourier modes being used in each direction. The output at each point is obtained via a projection layer in the form of a 2-layer multilayer perceptron (MLP) with width $(d_h, 128, d_u)$, where $d_u$ is the intended number of output. In the third example, $d_h$ is set to 16. A total of 3 Fourier layers with 32 Fourier modes in each direction are employed. The width of the projection MLP is set to $(d_h, 2d_h, d_u)$.

## B.1  Experiment 1 – Constitutive modeling of hyperelasticity

The dataset is obtained from Li et al. (2022a), which consists of an interpolated dataset of $41 \times 41$ point cloud on uniformly structured grids. The parameter of each method is given in the following, where the parameter choice of each model is selected by tuning the number of layers and the width (channel dimension) keeping the total number of parameters on the same magnitude.

- eDAFNO: In these cases, we use neural operators to construct mapping from grid location $\boldsymbol{x}$ as the input, and the stress field as the output. To perform fair comparison with the results reported in Li et al. (2022a), we employ the same hyperparameters here: in particular, four Fourier layers with mode 12 and width 32 are used.

- iDAFNO: The iDAFNO cases employ the same hyperparameters as the eDAFNO cases, with the iterative layer structure demonstrated in (11). In iDAFNO, all Fourier layers share the same set of trainable parameters, while different layers have different parameters in eDAFNO. Hence, iDAFNO reduces the number of trainable parameters by almost $75\%$, when using the same hyperparameters as in eDAFNO.

- FNO (with mask or smooth $\chi$): Following the same practice as in Li et al. (2022a), we train a plain FNO model (Li et al., 2020c), with the input as $[\boldsymbol{x}, \chi(\boldsymbol{x})]$ (in the "with mask" cases) or as $[\boldsymbol{x}, \tilde{\chi}(\boldsymbol{x})]$ (in the "with smoothed $\chi$" cases). Herein, we employ the same FNO architecture as reported in Li et al. (2022a), where four Fourier layers are used with mode 12 and width 32.

- IFNO (with mask or smooth $\chi$): Similar to the FNO cases, we also use $[\boldsymbol{x}, \chi(\boldsymbol{x})]$ (in the "with mask" cases) or $[\boldsymbol{x}, \tilde{\chi}(\boldsymbol{x})]$ (in the "with smoothed $\chi$" cases) as the input, with four Fourier layers, mode 12, and width 32. On the Fourier layers, the implicit architecture proposed in You et al. (2022c) is employed, such that all four Fourier layers share the same set of trainable parameters. Therefore, the number of trainable parameters in IFNOs is roughly 1/4 of that in FNOs.

- Geo-FNO: As a baseline model for FNOs with various geometries, we employ the Geo-FNO architecture from Li et al. (2022a), where an additional deformation neural network is trained together with FNO to provide a diffeomorphism from uniform grids to the deformed domain.

- F-FNO: Following the settings in Li et al. (2022a), we train the F-FNO model (Li et al., 2020c) with the input $[\boldsymbol{x}, \chi(\boldsymbol{x})]$. We adopt the same F-FNO architecture as reported in Tran et al. (2022), where four Fourier layers are used with mode 16 and width 64.

- GNO: The graph neural operators are flexible on the problem geometry, which have been widely used for complex geometries (Li et al., 2020a; Liu et al., 2022). To carry out fair comparison, we build a full graph with edge connection radius $r = 0.2$, width 32 and kernel width 512. As a result, the total number of parameters in GNOs is on the same magnitude as in FNOs.

- DeepONet: As another neural operator baseline model, the deep operator network (Lu et al., 2022) is composed of two neural networks – a trunk net and a branch net to represent the basis and coefficients of the operator. In this baseline, we use five layers for both the trunk net and branch net, each with a width of 256.

- UNet: Analogous to the setup in Li et al. (2022a), we train a UNet model (Ronneberger et al., 2015) on uniform grids, where 4 downsampling and upsampling blocks with 20 hidden channels are employed.

The comparison of the total number of parameters of the selected models used in the hyperelasticity problem is listed in Table 1[3]. In addition, the average runtime for each method on the hyperelasticity problem with 1000 training samples is provided in Table 4. All tests are performed on a NVIDIA RTX A6000 GPU card with 48GB memory. From this table, we can see that in DAFNOs the runtime increases slightly compared with the corresponding FNOs, but they are still substantially more efficient than other baselines, such as Geo-FNO.

For each method, we tune the learning rate from the range [1e-3,1e-1], the decay rate from the range [0.4,0.9], the weight decay parameter from from the range [1e-6,1e-2], and the smoothing coefficient (where applicable) from the range [5,100], then report the model with the best validation error. A typical training curve can be found in Figure 8. As a supplement of Table 2, the full table of all training and testing errors from different models is provided in Table 5.

Table 4: The per-epoch runtime (in seconds) of selected models for the hyperelasticity problem.

| model | eDAFNO | iDAFNO | FNO | IFNO | Geo-FNO | GNO | DeepONet | UNet | F-FNO |
|---|---|---|---|---|---|---|---|---|---|
| runtime | 2.00 | 1.70 | 1.81 | 1.62 | 5.12 | 98.37 | 940.12 | 5.04 | 3.41 |

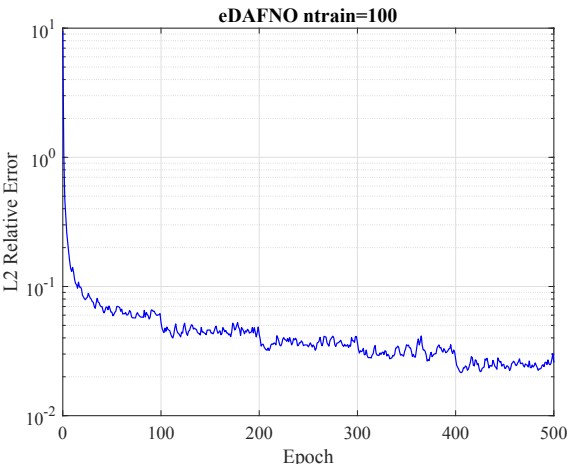

Figure 8: Demonstration of a typical training curve for eDAFNO.

To demonstrate the effect of the smoothing level when using different smoothing coefficient $\beta$, we illustrate the smoothed $\chi$ on an exemplar test sample in Figure 9. We also perform tests on the hyperelasticity example with a total of 1000 training samples and show the errors on the test dataset in Table 6. For each value of $\beta$, we search for the optimal initial learning rate, the decay rate, and the weight decay parameter based on the validation dataset, and report the optimal values.

---

[3]We note that the numbers of trainable parameters for the "Geo-FNO" and "FNO" cases are different from the ones provided in Li et al. (2022a). For fair comparison with methods using real-valued trainable parameters, we count each complex-valued trainable parameter as two degrees of freedom.

Table 5: Results for the hyperelasticity problem, where bold numbers highlight the best method according to the test error.

| Model, Dataset | # of training samples | | |
|---|---|---|---|
| | 10 | 100 | 1000 |
| eDAFNO, train | 6.800%±0.670% | 2.050%±0.035% | 0.664%±0.014% |
| eDAFNO, test | **16.446%±0.472%** | 4.247%±0.066% | **1.094%±0.012%** |
| iDAFNO, train | 7.266%±0.923% | 2.038%±0.036% | 0.812%±0.012% |
| iDAFNO, test | 16.669%±0.523% | **4.214%±0.058%** | 1.207%±0.006% |
| FNO w/ mask, train | 2.907%±0.318% | 2.277%±0.240% | 0.881%±0.015% |
| FNO w/ mask, test | 19.487%±0.633% | 7.852%±0.130% | 4.550%±0.062% |
| FNO w/ smooth $\chi$, train | 2.876%±0.152% | 2.058%±0.132% | 0.815%±0.012% |
| FNO w/ smooth $\chi$, test | 17.431%±0.536% | 5.479%±0.186% | 1.415%±0.025% |
| Geo-FNO, train | 0.547%±0.336% | 0.689%±0.676% | 1.192%±0.232% |
| Geo-FNO, test | 28.725%±2.600% | 10.343%±4.446% | 2.316%±0.283% |
| IFNO w/ mask, train | 2.274%±0.248% | 1.687%±0.047% | 2.701%±0.041% |
| IFNO w/ mask, test | 19.262%±0.376% | 7.700%±0.062% | 4.481%±0.022% |
| IFNO w/ smooth $\chi$, train | 3.704%±0.299% | 1.683%±0.029% | 1.013%±0.014% |
| IFNO w/ smooth $\chi$, test | 17.145%±0.432% | 5.088%±0.146% | 1.509%±0.018% |
| GNO, train | 27.337%±0.501% | 18.713%±0.669% | 13.321%±0.681% |
| GNO, test | 29.305%±0.321% | 18.574%±0.584% | 13.007%±0.729% |
| DeepONet, train | 23.071%±5.963% | 22.700%±0.984% | 7.937%±0.309% |
| DeepONet, test | 35.409%±0.408% | 25.925%±0.724% | 11.760%±0.827% |
| UNet, train | 98.042%±0.260% | 34.569%±2.676% | 1.760%±0.115% |
| UNet, test | 98.167%±0.236% | 34.467%±2.858% | 5.462%±0.048% |

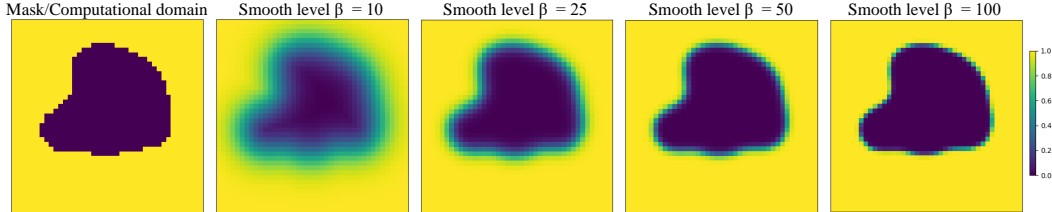

Figure 9: An illustration of the effect of varying the smoothing coefficient on the resulting boundary encoding. The larger the smoothing level $\beta$ is, the sharper and narrower the encoded boundary becomes. In effect, $\beta$ can be treated as a hyperparameter and tuned according to the validation error to either smoothen the boundary or keep the original boundary untouched.

Table 6: The effect of the smoothing coefficient $\beta$ on test loss in the hyperelasticity example with a total of 1000 training samples.

| initial learning rate | decay rate | weight decay parameter | $\beta$ | train loss | test loss |
|---|---|---|---|---|---|
| $4.5 \times 10^{-2}$ | 0.5 | $3 \times 10^{-6}$ | 5 | 0.564% | 1.155% |
| $4.0 \times 10^{-2}$ | 0.5 | $1 \times 10^{-5}$ | 10 | 0.637% | 1.064% |
| $2.0 \times 10^{-2}$ | 0.5 | $1 \times 10^{-5}$ | 20 | 0.454% | 1.120% |
| $1.5 \times 10^{-2}$ | 0.5 | $3 \times 10^{-5}$ | 30 | 0.516% | 1.147% |
| $2.5 \times 10^{-2}$ | 0.5 | $3 \times 10^{-5}$ | 40 | 0.608% | 1.135% |
| $1.5 \times 10^{-2}$ | 0.5 | $3 \times 10^{-5}$ | 50 | 0.504% | 1.179% |
| $1.5 \times 10^{-2}$ | 0.5 | $2 \times 10^{-5}$ | 60 | 0.498% | 1.194% |
| $1.5 \times 10^{-2}$ | 0.5 | $3 \times 10^{-5}$ | 70 | 0.508% | 1.240% |
| $1.5 \times 10^{-2}$ | 0.5 | $3 \times 10^{-5}$ | 80 | 0.515% | 1.275% |
| $1.5 \times 10^{-2}$ | 0.5 | $3 \times 10^{-5}$ | 90 | 0.529% | 1.306% |
| $3.0 \times 10^{-2}$ | 0.5 | $3 \times 10^{-5}$ | 100 | 0.675% | 1.338% |

## B.2  Experiment 2 – Airfoil design

The airfoil dataset is directly taken from Li et al. (2022a), which is an interpolated dataset of $101 \times 101$ point cloud on uniformly structured grids. The analytical mapping function $f$ and the corresponding inverse mapping function $f^{-1}$ used in Figure 5 are defined in the following:

$$\begin{bmatrix} x \\ y \end{bmatrix} = f\left(\begin{bmatrix} X \\ Y \end{bmatrix}\right) = \begin{bmatrix} 0.909 \tan^{-1}(1.965X) \\ 0.714 \tan^{-1}\left(3.46Y + 0.173 \sin\left(0.909\pi \tan^{-1}(1.965X)\right)\right) \end{bmatrix}, \tag{12}$$

$$\begin{bmatrix} X \\ Y \end{bmatrix} = f^{-1}\left(\begin{bmatrix} x \\ y \end{bmatrix}\right) = \begin{bmatrix} 0.509 \tan(1.1x) \\ 0.289 \tan(1.4y) - 0.05 \sin(\pi x) \end{bmatrix}, \tag{13}$$

where upper- and lower-case letters indicate the coordinate systems in the physical and computational spaces, respectively.

## B.3  Experiment 3 – Crack propagation with topology change

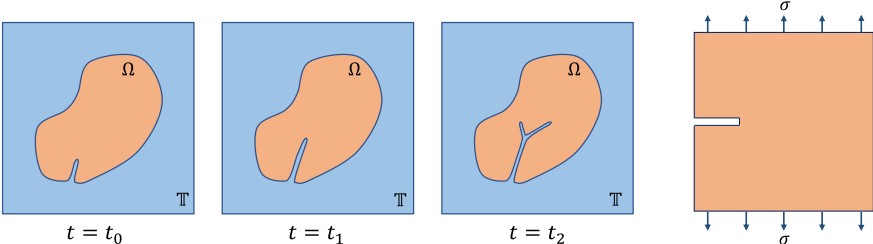

Figure 10: An illustration of the crack propagation problem, showing the topology change in DAFNO (left) and the physical problem setup (right), where a 2D plate with a pre-crack is subjected to external tractions (denoted as $\sigma$ with a slight abuse of notation) on the top and bottom edges.

An illustration of the time-dependent domain evolution, as well as the problem setup, is shown in Figure 10. The governing PD equation of motion for brittle fracture used in generating the dataset is given below:

$$\rho \frac{\partial^2 \boldsymbol{u}}{\partial t^2} = \mathcal{L}(\boldsymbol{u}) + \boldsymbol{b} \,, \; \mathcal{L}(\boldsymbol{u})(\boldsymbol{x}, t) = \int_{\mathcal{H}_x} \mu(\boldsymbol{x}, \boldsymbol{y}, t) f(\boldsymbol{x}, \boldsymbol{y}, t) d\boldsymbol{y} \,. \tag{14}$$

In (14), $\boldsymbol{u}$ is the displacement field, $\rho$ is mass density, $t$ is time, $\boldsymbol{b}$ is the external force density (a.k.a. the body force), and $\mathcal{L}(\boldsymbol{u})$ is the internal force density. $\mathcal{L}(\boldsymbol{u})$ is the divergence of stress in local theory, but in PD, it is defined by the integral described in (14). $\mathcal{H}_x$ denotes a finite-size neighborhood of point $\boldsymbol{x}$. $f(\boldsymbol{x}, \boldsymbol{y}, t)$ is the dual force density representing the pairwise force acting between unit volumes at points $\boldsymbol{x}$ and $\boldsymbol{y}$ in its neighborhood $\mathcal{H}_x$. $f$ depends on the PD constitutive model, and $\mu(\boldsymbol{x}, \boldsymbol{y}, t)$ is a binary history-dependent quantity representing material damage. $\mu$ is either 0 or 1 for brittle fracture models, where $\mu = 0$ denotes a lost interaction for material points $\boldsymbol{x}$ and $\boldsymbol{y}$ while $\mu = 1$ implies an intact connection between the two. For the material model, we choose to work with the linearized bond-based model, and for the damage model we adopt the pointwise energy-based model provided in the PeriFast software. According to the employed damage model, topology evolution due to growing crack is a function of strain energy which depends on the updated displacement field, i.e., $\Omega(t) = \Omega(\boldsymbol{u}(t))$. Additional details regarding the PD formulation can be found in Jafarzadeh et al. (2022b).

The physical parameters used in generating the data are: Young's modulus $E = 150$ GPa, Poison's ratio $\nu = 0.33$, mass density $\rho = 1000$ kg/m$^3$, and fracture energy $G_0 = 200$ J/m$^2$. The relative computational parameters are: PD horizon (the radius of the neighborhood for nonlocal interactions) $\delta = 2.07$ mm, extended domain (i.e., the periodic box) size $44.14$ mm $\times$ $44.14$ mm with $64 \times 64$ discretization, and time step $\Delta t = 2 \times 10^{-8}$ s. For the crack data subset in training, we run PeriFast software with the above parameters and the traction magnitude of 4 MPa. We record $u_1$, $u_2$, $\chi$, $L_1$, and $L_2$ for 450 consecutive time steps. For the sinusoidal data subset used in training, we set $u_1 = c \sin\left(\frac{2m\pi x_1}{L}\right) \sin\left(\frac{2n\pi x_2}{L}\right)$, and $u_2 = 0$ for $m, n = 1, 2, \cdots, 32$, where $L$ is the length of the square box, $x_1$ and $x_2$ are the 2D coordinates, and $c = 0.01/32$ is a scaling factor to make

the generated displacement in the same scale as the crack data. Additionally, we set $u_1 = 0$ and $u_2 = c \sin\left(\frac{2m\pi x_1}{L}\right) \sin\left(\frac{2n\pi x_2}{L}\right)$. This results in a total of 2,048 instances of sinusoidal displacement fields. Next, we set $\chi = 1$ for all nodes and use the PD operator in PeriFast to compute the corresponding $L_1$ and $L_2$ fields.

Following the common practice in PD simulations (Ha & Bobaru, 2010), we employ the following two additional techniques to help with training and stabilizing crack propagation. Firstly, we do not allow damage to initiate from boundaries. This technique has been used in previous PD simulations and is referred to as the "no-fail zone". It effectively stops unrealistic distributed damage from initiating on the boundaries. Secondly, given that the physical problem is symmetric, we enforce the damage growth in the simulation with eDAFNO to be symmetric as well. Note that the whole domain is used for training, and the predictions on the entire domain is used for next time step evaluation. Symmetry is enforced only when the topology characteristic function $\chi$ is updated.

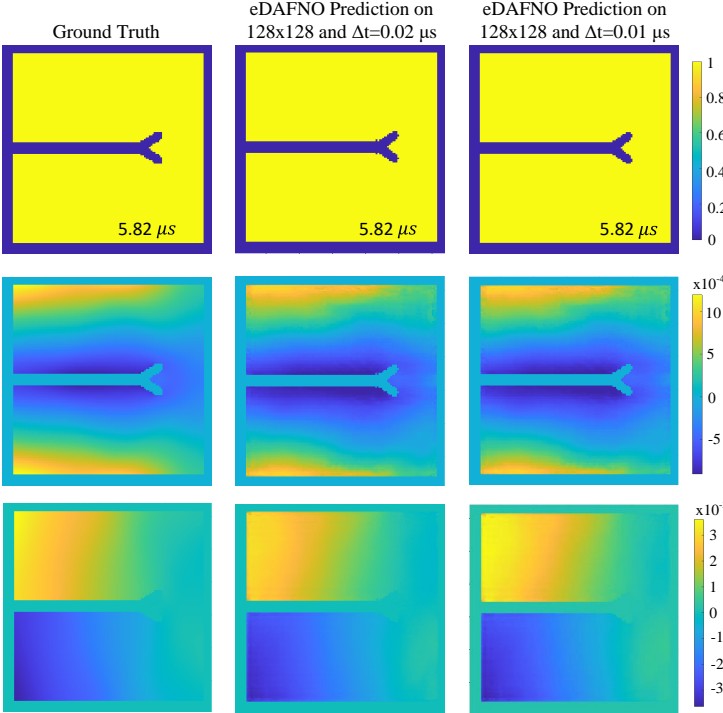

Figure 11: Demonstration of the resolution-independence property of eDAFNO trained using a spatial discretization of $64 \times 64$ and time step of 0.02 $\mu$s and tested on a spatial discretization of $128 \times 128$ and time step of 0.01 $\mu$s. The three rows correspond to the $\chi$, $u_1$, and $u_2$ fields, respectively.

Besides the resolution-independence property of DAFNO as shown in Figure 3, we further investigate the generalizability of DAFNO in both physical and temporal resolutions with this example. Specifically, the eDAFNO model is trained on a spatial resolution of 64×64 and a time step of 0.02 $\mu$s, and it is here tested on both a finer spatial resolution of 128×128 and a finer time step of 0.01 $\mu$s. As shown in Figure 11, the performance of the low-resolution-trained eDAFNO on high resolutions is compared with the high-fidelity peridynamics simulation results, where visually identical results are observed. Note that, although the time marching is computed with an ODE solver in this example, the temporal resolution independence is still worth investigating because, as the number of time steps increases, the number of times that the error accumulates in the dynamic solver increases as well. Our results show that eDAFNO prediction remains independent of the time step employed.

### B.4 Experiment 4 – Pipe flow

We perform an additional experiment of pipe flow, in which the dataset is obtained from Li et al. (2022a). We closely follow their problem setup and briefly document the comparison against Geo-FNO in what follows.

Using 1000 samples for training, eDAFNO has achieved a similar performance to Geo-FNO: when comparing the relative $L^2$ errors, eDAFNO's test error on the pipe dataset is 0.71%, while the test error of Geo-FNO is 0.67%. When comparing the maximum absolute error (cf. Figure 12), eDAFNO has $0.051$, while Geo-FNO has $0.061$. This is probably due to the fact that all pipes have a very simple geometry, which can be accurately represented with the pre-specified mapping in Geo-FNO. Note that such a pre-specified mapping for grid deformation can be easily added in DAFNO, as demonstrated in the airfoil experiment. In this circumstance, DAFNO becomes exactly the same as Geo-FNO in the pipe flow setup.

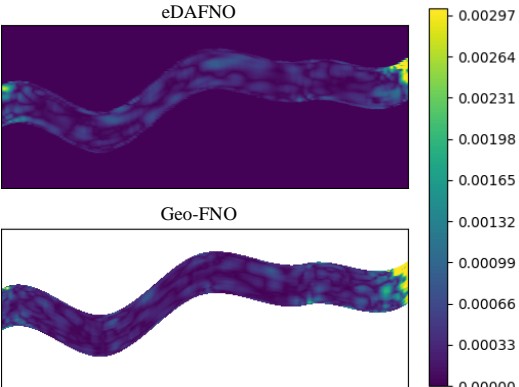

Figure 12: An illustration of the absolute error distribution of predictions from Geo-FNO and eDAFNO on the pipe dataset. The maximum absolute errors of Geo-FNO and eDAFNO are 0.061 and 0.051, respectively. In the vicinity of the outlet where most errors accumulate, eDAFNO is also more accurate compared to Geo-FNO.