# OpenReview forum: "Domain Agnostic Fourier Neural Operators"
_NeurIPS.cc/2023/Conference — NeurIPS 2023 poster_

### Official Review · Reviewer_BzBv · 2023-06-15

**Soundness:** 3 good
**Presentation:** 3 good
**Contribution:** 3 good
**Rating:** 5
**Confidence:** 5

**Summary:**

This paper presents a novel method of extending Fourier Neural Operators (FNO) to irregular geometries using an indicator function to represent the shape of the geometric area. This area is then extended to a larger, regular area, thereby allowing FNO to handle irregular geometries. This is indeed an innovative approach, as the use of indicator functions in neural operators to handle irregular areas has not been done before. The methodology bears some similarities to the use of neural networks in  PINNs to learn hard constraints, ensuring regular equations comply with constraints on the geometric boundary. Experimental results indicate that the proposed method achieves commendable performance with fewer parameters.

**Strengths:**

The paper's idea is innovative, using indicator functions in neural operators to handle irregular geometries is a novel approach. The authors show through experimental results that the proposed method performs well with fewer parameters, a significant advantage.

**Weaknesses:**

1. The review of related work seems insufficient. The recent development of methods represented by transformers [1,2,3] that handle irregular geometric areas well is not mentioned or cited at all. Moreover, last year's ICLR Factorized FNO [4] also dealt with several non-uniform geometric area problems.
2. Regarding the technical aspect of the paper, I have a concern. The original irregular grid may be adaptive, implying that we may need a very high resolution when embedding it into a uniform grid. This might mean that the uniform grid's spacing needs to be smaller than the smallest spacing of the original grid, leading to considerable computation waste. The proposed method does not seem to avoid this waste.
3. I suggest that for datasets like airfoil and hyperelasticity, which have been detailed in other works, there is no need for further extensive description in the main text. From a machine learning researcher's perspective, the dataset's attributes, sizes, and challenges are more important than detailed principles of its generation, especially when these have already been thoroughly discussed in other studies.

Considering the above points, the authors need first add a few key references, carefully consider the technical questions raised, and then heavily revise the paper. If the core technical challenges (question 2 ) can be addressed, I may consider increasing the score after the rebuttal. Otherwise, I would suggest that the authors continue to revise this work for submission to another conference or journal.

1. Choose a Transformer: Fourier or Galerkin (https://arxiv.org/abs/2105.14995)
2. Transformer for Partial Differential Equations' Operator Learning (https://arxiv.org/abs/2205.13671)
3. GNOT: A General Neural Operator Transformer for Operator Learning (https://arxiv.org/abs/2302.14376)
4. Factorized Fourier Neural Operators (https://arxiv.org/abs/2111.13802)

**Questions:**

None.

**Limitations:**

None.

---

> ### Author Rebuttal · Authors · 2023-08-10
>
> We thank the reviewer for their constructive comments.
>
> **References on transformer-type neural operators and F-FNO**: We thank the reviewer's kind suggestions, and have added F-FNO [4] as a baseline in our comparison. As shown in the tables below: in both the hyperelasticity and airfoil design problems, DAFNO outperforms F-FNO.  Regarding the transformer-type neural operators, we would like to point out that [1] is based on regular grids and does not handle irregular geometric areas. While [2] and [3] can handle irregular geometric areas, they got officially published in April and July this year, respectively, which were around the same time this manuscript was submitted. We will cite these work in our revised paper.
>
> **Handling non-uniform grids**: DAFNO can be easily combined with Geo-FNO and include either an analytical or trainable mapping from non-uniform/irregular mesh grids to uniform mesh grids. As a demonstration of this capability, we consider irregular grids in the airfoil problem and use a pre-computed function to map them to regular grids, as shown in Fig. 1 in the attached pdf file of the global response. Then, we train eDAFNO on regular grids and map the results back. The test loss on irregular grids using the eDAFNO learned model is 0.659\%$\pm$0.007\%, which is similar to the DAFNO results on uniform grids.
>
> Test errors for the hyperelasticity problem, where bold numbers highlight the best method.
>
> | Model| 10 samples | 100 samples | 1000 samples |
> | :------------- | :-----------: | :-----------: | :-----------: |
> |eDAFNO | **16.446\%**$\pm$**0.472\%** | 4.247\%$\pm$0.066\% | **1.094\%**$\pm$**0.012\%**|
> |iDAFNO | 16.669\%$\pm$0.523\% | **4.214\%**$\pm$**0.058\%** | 1.207\%$\pm$0.006\%|
> |FNO w/ mask | 19.487\%$\pm$0.633\% | 7.852\%$\pm$0.130\% | 4.550\%$\pm$0.062\% |
> |IFNO w/ mask | 19.262\%$\pm$0.376\% | 7.700\%$\pm$0.062\% | 4.481\%$\pm$0.022\% |
> |Geo-FNO | 28.725\%$\pm$2.600\% | 10.343\%$\pm$4.446\% | 2.316\%$\pm$0.283\% |
> |GNO | 29.305\%$\pm$0.321\% | 18.574\%$\pm$0.584\% | 13.007\%$\pm$0.729\% |
> |DeepONet | 35.334\%$\pm$0.179\% | 25.455\%$\pm$0.245\% | 11.998\%$\pm$0.786\%|
> |F-FNO | 35.672\%$\pm$3.852\% | 12.135\%$\pm$5.813\% | 3.193\%$\pm$1.622\%|
> |UNet | 98.167\%$\pm$0.236\% | 34.467\%$\pm$2.858\% | 5.462\%$\pm$0.048\%|
> |FNO w/ smooth $\chi$ | 17.431\%$\pm$0.536\% | 5.479\%$\pm$0.186\% | 1.415\%$\pm$0.025\% |
> |IFNO w/ smooth $\chi$ | 17.145\%$\pm$0.432\% | 5.088\%$\pm$0.146\% | 1.509\%$\pm$0.018\% |
>
> Test errors for the airfoil problem.
>
> | Model | Train error | Test error |
> | :------------- | :-----------: | :-----------: |
> |eDAFNO | 0.329\%$\pm$0.020\% | **0.596\%**$\pm$**0.005\%** |
> |iDAFNO | 0.448\%$\pm$0.012\% | 0.642\%$\pm$0.020\% |
> |eDAFNO on irregular grids | 0.331\%$\pm$0.003\% | 0.659\%$\pm$0.007\% |
> |Geo-FNO | 1.565\%$\pm$0.180\% | 1.650\%$\pm$0.175\% |
> |F-FNO | 0.566\%$\pm$0.066\% | 0.794\%$\pm$0.025\% |
> |FNO w/ mask | 2.676\%$\pm$0.054\% | 3.725\%$\pm$0.108\% |
> |UNet w/ mask | 2.781\%$\pm$1.084\% | 4.957\%$\pm$0.059\% |

---

> > ### Comment · Reviewer_BzBv · 2023-08-22
> > **Feedback**
> >
> > After reading your rebuttal, most of my concerns were resolved. I have raised my score to 5, but not higher due to the concern that I still think computing grid transformations to a uniform grid (like Geo-FNO) is inefficient and has many limitations. By reading your revision I found this is still a challenge. For example, for 3D airfoil problems, you will need very high-resolution grids which might be computationally expensive.

---

### Official Review · Reviewer_dnax · 2023-07-04

**Soundness:** 3 good
**Presentation:** 3 good
**Contribution:** 3 good
**Rating:** 6
**Confidence:** 3

**Summary:**

The paper explores the extension of Fourier Neural Operators (FNOs) to irregular geometries and topology changes. To leverage the computational speed benefits of the fast Fourier transform (FFT) employed by FNOs, they enclose the physical domain with a period box. They then adapt the FNO kernel by multiplying the integrand with a smoothed characteristic function that encodes domain information. This encoding enables their method to generalize to new geometries while preserving the FFT computational efficiency. They outperform several FNO baselines on two benchmark datasets, and showcase the properties of their method on a real-world crack propagation dataset, which contains evolving topology.

**Strengths:**

The paper is well written and clear. The different choices of architecture are well justified.
- It tackles an interesting and useful problem for real-world application of FNO.
- The idea of incorporating the domain information in the kernel is simple, but effective and grounded, and the smoothing of the domain characteristic function helps avoiding dicontinituty problems.
- Experimental results on three datasets are strong.
- The crack propagation dataset is well-considered and helps highlighting the advantages of DAFNO, especially its ability to adapt to evolving topologies.

**Weaknesses:**

- The comparison with several baselines is interesting, but given the recent improvements seen over FNO and Geo-FNO, having F-FNO [1] as a baseline seems important. In addition to neural operators, other grid-independent methods that can be used on irregular domains have been developed, for instance [2] implemented a continuous model using Implicit Neural Representations (INRs). It would have been interesting to see a comparison or at least a discussion about the advantage of DAFNO in the light of these recent improvements in the community (for e.g. computational cost?).

- As explained in the limitations and clear from the methodology itself, the architecture struggles to handle non-uniformly meshed grids due to the inability to use FFT on such grids. This limitation significantly restricts the range of problems that can be efficiently tackled using this architecture. For example, if applied to real-world airfoil surrogate modeling problems like [3], it seems that DAFNO would not offer significant improvements over the original FNO.

- While the idea behind DAFNO is interesting and novel, the architectural improvement upon the regular FNO architecture seems limited. The authors address this issue in a ‘remark’ paragraph, but I am not totally convinced that it is a sufficient improvement, especially since the architecture does not appear to efficiently handle non-uniformly meshed grids.

- This is more a remark than a weakness. In Figure 3, the zero-shot super-resolution prediction from eDAFNO does not effectively highlights the resolution invariant properties, since there is no ground truth to compare it to. While some degree of "deblurring" is noticeable, it remains unclear if the network successfully captures higher frequency phenomena that may occur at higher resolutions. The appendix, specifically Figure 10, sheds light on this property for the crack propagation dataset. However, Figure 3 does not truly demonstrates this resolution invariant property.

[1]: Tran, Alasdair, et al. "Factorized Fourier Neural Operators." The Eleventh International Conference on Learning Representations. 2022.

[2]: Yin, Yuan, et al. "Continuous PDE Dynamics Forecasting with Implicit Neural Representations." The Eleventh International Conference on Learning Representations. 2023.

[3]: Bonnet, Florent, et al. "AirfRANS: High Fidelity Computational Fluid Dynamics Dataset for Approximating Reynolds-Averaged Navier–Stokes Solutions." Advances in Neural Information Processing Systems 35 (2022): 23463-23478.

**Questions:**

1. I wonder why Geo-FNO is not in the baselines for the Crack propagation dataset. It would have been an interesting baseline, especially to see if it could adapt to evolving geometries better than a masked FNO. It is claimed that Geo-FNO cannot handle topology change, but once a continuous mapping is learned, it can be applied. I agree that it should not adapt well, but is there a specific reason for its absence in these experimental results?

2. Just to confirm my understanding, for the FNO baseline, the grid is interpolated as in Geo-FNO, and then a mask is given as an additional input?

3. I don’t quite understand the link from equation 5 to the equation before 6. Why is $\chi(x)$ in factor of $W^l h(x) + c^l$ ? Is it to ensure that DAFNO will not produce results for points outside of the domain?

4. Given the link of the smoothing function to a signed distance function (SDF) with the $\\tanh(\\beta \\text{dist}(x, \partial \Omega))$ formulation, is there a reason for not directly using a SDF instead of the smoothing function ? It may have given more (or a different) information to the network, as in different previous works (for e.g. [4]).

5. Out of curiosity, have you experimented with using other smoothing functions besides tanh (for e.g. a logistic function)?

[4]: Guo, Xiaoxiao, Wei Li, and Francesco Iorio. "Convolutional neural networks for steady flow approximation." Proceedings of the 22nd ACM SIGKDD international conference on knowledge discovery and data mining. 2016.

**Limitations:**

The main limitations are already presented in the article.

---

> ### Author Rebuttal · Authors · 2023-08-10
>
> We thank the reviewer for their valuable suggestions.
>
> **Comparison with additional baselines**: We have added F-FNO as an additional baseline in both the elasticity and airfoil problems. On the other hand, since INR focuses on learning a time-continuous dynamics model of the underlying flow, and it is not applicable to these two baseline problems. We would like to point out that when comparing F-FNOs with DAFNOs, our original conclusions still stand: DAFNOs consistently outperform F-FNOs in accuracy, with halved computational time.
>
> Test errors for the hyperelasticity problem.
>
> | Model| 10 samples | 100 samples | 1000 samples |
> | :------------- | :-----------: | :-----------: | :-----------: |
> |eDAFNO | **16.446\%**$\pm$**0.472\%** | 4.247\%$\pm$0.066\% | **1.094\%**$\pm$**0.012\%**|
> |iDAFNO | 16.669\%$\pm$0.523\% | **4.214\%**$\pm$**0.058\%** | 1.207\%$\pm$0.006\%|
> |Geo-FNO | 28.725\%$\pm$2.600\% | 10.343\%$\pm$4.446\% | 2.316\%$\pm$0.283\% |
> |F-FNO | 35.672\%$\pm$3.852\% | 12.135\%$\pm$5.813\% | 3.193\%$\pm$1.622\%|
>
> The per-epoch runtime (second) in the hyperelasticity problem.
>
> |Model | eDAFNO | iDAFNO | FNO | IFNO | Geo-FNO | F-FNO |
> | :------------- | :-----------: | :-----------: | :-----------: | :-----------: | :-----------: | :-----------: |
> |runtime (s) | 2.00 | 1.70 | 1.81 | 1.62 | 5.12 | 3.41 |
>
> Test errors for the airfoil problem.
>
> | Model | Train error | Test error |
> | :------------- | :-----------: | :-----------: |
> |eDAFNO | 0.329\%$\pm$0.020\% | **0.596\%**$\pm$**0.005\%** |
> |iDAFNO | 0.448\%$\pm$0.012\% | 0.642\%$\pm$0.020\% |
> |eDAFNO on irregular grids | 0.331\%$\pm$0.003\% | 0.659\%$\pm$0.007\% |
> |Geo-FNO | 1.565\%$\pm$0.180\% | 1.650\%$\pm$0.175\% |
> |F-FNO | 0.566\%$\pm$0.066\% | 0.794\%$\pm$0.025\% |
>
> **Handling non-uniform grids**: We would like to point out that DAFNO can be readily combined with the grid mapping technique in Geo-FNO, to handle non-uniform grids. As a demonstration of this capability, we consider irregular grids in the airfoil problem and use a pre-computed function to map irregular grids to regular grids, then we train eDAFNO. In this irregular grid set, we place more grid points near the airfoil, so as to provide a better resolution near the important parts as suggested by the reviewer. The test loss using the eDAFNO learned model is 0.659\%$\pm$0.007\%, which is similar to the DAFNO results on uniform grids. In Fig. 1 of the attached pdf file in the global response we demonstrate the irregular mesh, and in Fig. 2 we plot the errors of Geo-FNO, DAFNO on regular grids, and DAFNO on irregular grids. One can see that while both DAFNOs substantially outperform Geo-FNO, the error contours from eDAFNO with irregular mesh show a smaller miss-match region near the airfoil, verifying the flexibility of DAFNO in meshing and its capability in resolving possible fine-grained feature in real-world modeling problems.
>
> The original paper focused on uniform grids, just to highlight our simple but efficient architecture to embed characteristic domain encoding -- which can be readily combined with the grid deformation technique in Geo-FNO to handle non-uniform grids, in addition to its unique capability in handling topological changes.
>
> **Geo-FNO for crack propagation**: The crack example is presented intentionally as a problem where Geo-FNO falls short and cannot be used. The reason is that the Geo-FNO requires a pre-defined or trainable isomorphism between a rectangular domain and the targeted irregular domains. However, in fracture problems the domain undergoes severe topological changes including the emergence of new holes or discontinuities/boundaries (as shown in Fig. 5) which is not known a priori. Such an isomorphism does not exist (intuitively, there is no way to define a mapping between each yellow regions in Fig. 5 to a rectangular domain).
>
> **FNO baseline**: The reviewer is correct that the grid in the FNO baseline is interpolated as in Geo-FNO, and then a mask is given. The settings as well as the interpolated benchmark datasets are consistent with the Geo-FNO paper.
>
> **Explanation of $\chi(x)$ in factor of $W^l(x)+c^l$**: Yes, the reviewer is correct. Adding $\chi(x)$ in factor of $W^l(x)+c^l$ makes the values outside of the main domain zero. Additionally, it allows us to factor out $\chi(x)$ from the entire equation and present it in the form of Eq. 6.
>
> **Using SDF as the characterization function**: We didn't use signed-distance function directly because there is no physical meaning: the characterization function $\chi$ in our setting is a smoothed binary representation which aims to provide near-local structural information. By multiplying it in the integral layer, we can eliminate the interactions between points inside and outside the physical domain, similar as in the molecular dynamics method [1]. As pointed out by [2], SDF aims to provide global structural information and its effect is very different from the local information from binary representation. To them, SDF works but binary representation does not, probably due to the fact that their NN architecture is substantially different from ours: SDF was used as an input, while the characteristic function was applied to layer weights in our setting. Moreover, the tanh on distance function aims to provide a smoothing function, while the SDF itself is not smooth.
>
> [1] Hansson, Tomas et al. "Molecular dynamics simulations." Current opinion in structural biology 12.2 (2002): 190-196.
>
> [2] Guo, Xiaoxiao et al. "Convolutional neural networks for steady flow approximation." Proceedings of the 22nd ACM SIGKDD international conference on knowledge discovery and data mining. 2016.
>
> **Other smoothing functions**: We have also tested other smoothing functions, such as the Gaussian filter. The effect is very similar to tanh. This fact was also verified from Table 6 in our ablation study: As far as the characteristic function has a sufficient level of smoothness, the test error does not vary much.

---

> > ### Comment · Reviewer_dnax · 2023-08-17
> > **Thank you for your rebuttal**
> >
> > Thank you for all your differents answers, which I found insightful. My two main reservations concerned the limited related work missing the F-FNO reference, as well as the potential inapplicability of the architecture to non-uniformly meshed grids. The authors have effectively addressed both of these concerns, consequently I will improve my evaluation.

---

> > > ### Author Response · Authors · 2023-08-19
> > > **Thank you!**
> > >
> > > We thank the reviewer for their kind response and for raising the score. We sincerely appreciate the reviewer's valuable time and suggestions, which helped us to improve the quality of this work.

---

### Official Review · Reviewer_UPhM · 2023-07-05

**Soundness:** 2 fair
**Presentation:** 2 fair
**Contribution:** 2 fair
**Rating:** 6
**Confidence:** 3

**Summary:**

The Fourier Neural Operator (FNO) is a model in the field of neural operators that has successfully interpreted various physical phenomena. However, one of the issues with FNO is its limitation to learn only on rectangular domains. In Geo-FNO, this problem is addressed by lifting irregular domains to the latent space of rectangular domains.

The authors propose a novel convolution integral kernel through the domain characteristic function X(x) and introduce DAFNO (Domazin-Adapted Fourier Neural Operator). DAFNO outperforms the existing baseline in problems involving airfoil and hyperelasticity.


**Strengths:**

Conducting research on a domain-agnostic physics simulator is a crucial endeavor. In particular, the field of neural operators, which the authors have explored, offers a highly efficient approach zto interpreting recent physical information, thus holding significant potential.

**Weaknesses:**

Could you provide more detailed explanation about the mathematical motivation behind the domain characteristic function? It would be necessary to provide specific explanations for Eq(4) and Eq(5).

Due to the varying model sizes between the baseline models and the proposed model in Table 1, it is difficult to claim that the comparison is fair. A more equitable comparison can be achieved when all model sizes are standardized to be the same.

The authors proposed a new neural operator model called DAFNO. However, besides modifying the internal structure of the iterative layers in Geo-FNO, specifically by splitting the structure of the iterative layers in Figure 2, it is challenging to identify any other distinct novelties.

Adding more baselines in the field of neural operators could lead to better experiments and improved evaluations. Such as, MWT (Multiwavelet-based neural operator), WNO (Wavelet neural operator)


**Questions:**

Is there no experimental result for the benchmark dataset (Pipe) based on the Navier-Stokes equation in the Geo-FNO paper? I regard it as one of the significant experiments in this domain.

Intuitively, it might be expected that DAFNO, having designed the Iterative layers in parallel, could have larger training and inference times compared to the baseline models like FNO and Geo-FNO. Can you provide any measurement results to demonstrate this?


**Limitations:**

There is no limitations.

---

> ### Author Rebuttal · Authors · 2023-08-10
>
> We thank the reviewer for their constructive comments.
>
> **Mathematical motivation behind the domain characteristic function**: In order to obtain a truly domain-independent operator, we aim to hard-code domain information into the architecture. The challenge is to maintain the applicability of FFT while hard-coding the bounded, arbitrarily shaped domain. The aim is achieved if we enclose the bounded domain inside the rectangular box (so FFT is applicable), but "cut" the interactions between the main domain and the exterior, as if they are two non-interacting bodies. This means that when FNO integral is being computed for a point $x$, integration over points $y$ that fall onto the exterior should be avoided. This is achieve by adding $\chi(x)\chi(y)$ inside the integral. $\chi$ is 1 inside and 0 outside of the domain; as a result, the integrand becomes zero for $(x,y)$ pairs that one falls inside and one falls outside of the domain since $\chi(x)\chi(y) = 0$ for such pairs. Hence, the main domain and the exterior are separated. With this simple but effective modification, the integral operator is still defined over a box and the convolution form of the integral is preserved. As a result, FFT remains applicable while the bounded domain info is hard-coded into the architecture.
>
> **Unify model sizes in table 1**: We have updated the total number of parameters in DeepONet to be on the same level as other models ($\sim$2.5M), and the results are updated accordingly. Here, we point out that the only exceptions are the two implicit models (i.e., iDAFNO and IFNO), which have much fewer parameters. This is because these models are designed to be layer-independent in parameters. In other words, different layers share the same set of parameters, and therefore the total number of parameters stays the same with the increase of layers. To provide a fair comparison between explicit and implicit models, we employ the same architectural hyperparameters (number of layers, width, modes, etc), for eDAFNO, iDAFNO, FNO, and IFNO, while keeping the number of parameters in all explicit models on the same level as other baselines. As a result, implicit models having far fewer parameters than others.
>
> The total number of parameters and per-epoch runtime (second) in the hyperelasticity problem.
>
> |Model | eDAFNO | iDAFNO | FNO | IFNO | Geo-FNO | GNO | DeepONet | UNet | F-FNO |
> | :------------- | :-----------: | :-----------: | :-----------: | :-----------: | :-----------: | :-----------: | :-----------: | :-----------: | :-----------: |
> |nparams | 2.37M | 0.60M | 2.37M | 0.60M | 3.02M | 2.64M | 3.10M | 3.03M | 3.21M|
> |runtime (s) | 2.00 | 1.70 | 1.81 | 1.62 | 5.12 | 98.37 | 940.12 | 5.04 | 3.41 |
>
> Test errors for the hyperelasticity problem.
>
> | Model| 10 samples | 100 samples | 1000 samples |
> | :------------- | :-----------: | :-----------: | :-----------: |
> |eDAFNO | **16.446\%**$\pm$**0.472\%** | 4.247\%$\pm$0.066\% | **1.094\%**$\pm$**0.012\%**|
> |iDAFNO | 16.669\%$\pm$0.523\% | **4.214\%**$\pm$**0.058\%** | 1.207\%$\pm$0.006\%|
> |FNO w/ mask | 19.487\%$\pm$0.633\% | 7.852\%$\pm$0.130\% | 4.550\%$\pm$0.062\% |
> |IFNO w/ mask | 19.262\%$\pm$0.376\% | 7.700\%$\pm$0.062\% | 4.481\%$\pm$0.022\% |
> |Geo-FNO | 28.725\%$\pm$2.600\% | 10.343\%$\pm$4.446\% | 2.316\%$\pm$0.283\% |
> |GNO | 29.305\%$\pm$0.321\% | 18.574\%$\pm$0.584\% | 13.007\%$\pm$0.729\% |
> |DeepONet | 35.334\%$\pm$0.179\% | 25.455\%$\pm$0.245\% | 11.998\%$\pm$0.786\%|
> |F-FNO | 35.672\%$\pm$3.852\% | 12.135\%$\pm$5.813\% | 3.193\%$\pm$1.622\%|
> |UNet | 98.167\%$\pm$0.236\% | 34.467\%$\pm$2.858\% | 5.462\%$\pm$0.048\%|
>
> **Challenging to identify distinct novelties**: To our best knowledge (and also as pointed out by reviewers FUid and BzBv), our work has for the first time proposed to modify the internal structure of the iterative layers and encode domain geometry. As a result, DAFNOs are the first neural operator that can represent and handle dynamically changing domain topology (also pointed out by reviewer FUid).
>
> **Adding more baselines such as MWT**: We appreciate the reviewer's suggestion. The reason why we did not use them as baselines in our current work is that MWT requires the input grids to be 2 to the integer powers (i.e., $2^N$) and does not fit the benchmark datasets. We will include a discussion in the revised manuscript to acknowledge the contribution of MWT and MNO. Per the reviewer's suggestion, we have added Factorized FNO (F-FNO) as an additional baseline in both the elasticity and airfoil problems (see the table for elasticity problem above, and results of airfoil problem in global response), and the original findings still stand: DAFNOs consistently outperform all baselines.
>
> **Baseline on pipe flow example from Geo-FNO**: Per suggested by the reviewer, we have added an additional test of DAFNO on the pipe flow example. As shown in Fig.  3 of the attached pdf file in the global response, eDAFNO has achieved a similar performance to Geo-FNO: when comparing the relative $L^2$ errors, eDAFNO's test error on the pipe dataset is 0.719\%, while the test error of Geo-FNO is 0.67\%. When comparing the maximum absolute error, eDAFNO has 0.051, while Geo-FNO has 0.061. This is probably due to the fact that all pipes have a very simple geometry, which can be accurately represented with the pre-specified mapping in Geo-FNO. We want to further comment that such a pre-specified mapping for grid deformation can also be readily added to DAFNO, as demonstrated in Fig. 1 in the attached pdf file of the global response for the airfoil problem. In that case, DAFNO will be exactly the same as Geo-FNO.
>
> **Computational cost comparison**: We have already reported the comparison in terms of the runtime in Table 4 in the Appendix (which is also copied above). As demonstrated in the table, DAFNOs' runtime is similar to FNO, and beats Geo-FNO by 67\% in iDAFNO and 61\% in eDAFNO.

---

> > ### Comment · Reviewer_UPhM · 2023-08-18
> >
> > Thank you for answering all my concerns and questions. Reflecting this, I will raise my score (Border line reject to Weak accept).

---

> > > ### Author Response · Authors · 2023-08-19
> > > **Thank you!**
> > >
> > > We thank the reviewer for their kind response and for raising the score. We sincerely appreciate the reviewer's valuable time and suggestions, which helped us to improve the quality of this work.

---

### Official Review · Reviewer_fNa3 · 2023-07-06

**Soundness:** 3 good
**Presentation:** 3 good
**Contribution:** 3 good
**Rating:** 7
**Confidence:** 4

**Summary:**

In this paper, the authors propose the Domain Agnostic Fourier Neural Operator (DAFNO), an FNO that can deal with irregular boundaries. While the classical FNO is limited by construction to irregular domains, DAFNO simply includes a smoothed function $I(\cdot)$  of the characteristic function of the domain on which the data is defined. DAFNO is shown to outperform relevant baselines on 2-dimensional problems with irregular boundaries and one problem with topology changing over time.

**Strengths:**

The paper is about a very simple yet effective idea: including a smoothed mask to model boundaries inside the integral operator. This approach is original to my knowledge and effective in terms of strong experimental evaluation relevant to practical applications, including being able to handle topology changes. The paper is clear in its description and well-placed in the current literature on neural operators and deep surrogate models for scientific ML.

**Weaknesses:**

In my opinion, there is no major weakness in this paper. The main limitation could be that the approach is (particularly) simple, but I believe this does not need to be seen as a negative point. Sometimes, simpler is better. The experimental results are solid but do not encompass more complex settings such as 3D fluid dynamics. Another weakness is the lack of source code, which I believe should be provided, given the simplicity of the setting.

**Questions:**

- Can the proposed method be extended to modeling systems ≥ than 3 dimensions?
- At line 262: “In general, the topology evolution rule that determines Ω(t) can be obtained from a separate neural network or from physics as is the case in the current example”, did you test out a separate neural network in the experiments as well?

**Limitations:**

The main limitation (about uniformly meshed grids) is mentioned. Other limitations are covered in the “weaknesses” section.

---

> ### Author Rebuttal · Authors · 2023-08-10
>
> We thank the reviewer for their valuable suggestions.
>
> **High-dimensional problems**: DAFNOs are readily applicable to more complex settings and higher-dimensional problems, as neither the characteristic geometric encoding nor the smoothening technique is constrained to a specific dimension. It will just require more memory when it comes to higher dimensions.
>
> **Release code**: We have uploaded our DAFNO code of the hyperelasticity problem on anonymous github and sent the link to the Area Chair. Our DAFNO package on other problems will also be made publicly available on github once the paper is accepted.
>
> **Topology evolution rule as NN**: No yet. Practically, one can calculate the topology evolution rule in the form of fracture energy [1] or damage field [2] using a separate neural network, although that might introduce additional modeling error in long-term propagation. In this work we used physical laws so as to focus on resolving the geometric changes.
>
> [1] Goswami, Somdatta, et al. "A physics-informed variational DeepONet for predicting crack path in quasi-brittle materials." Computer Methods in Applied Mechanics and Engineering 391 (2022): 114587.
>
> [2] You, Huaiqian, et al. "Learning deep implicit Fourier neural operators (IFNOs) with applications to heterogeneous material modeling." Computer Methods in Applied Mechanics and Engineering 398 (2022): 115296.

---

> > ### Comment · Reviewer_fNa3 · 2023-08-19
> > **Thanks**
> >
> > Thanks for your reply.
> > (The code has not been shared with us reviewers, but I trust the AC will check it.)
> > Given my concerns have mostly been solved and the response to the other reviewers, I am happy to raise my score!

---

> > > ### Author Response · Authors · 2023-08-19
> > > **Thank you!**
> > >
> > > We thank the reviewer for their kind response and for raising the score. We sincerely appreciate the reviewer's valuable time and suggestions, which helped us to improve the quality of this work.

---

### Official Review · Reviewer_FUid · 2023-07-10

**Soundness:** 4 excellent
**Presentation:** 4 excellent
**Contribution:** 3 good
**Rating:** 6
**Confidence:** 4

**Summary:**

Fourier Neural Operators (FNOs) are a popular method for modeling physical systems such as different types of PDEs. However, in order to use the FFT to make FNO efficient, the input needs to be a regular grid. The authors study the question of irregular grid inputs for FNO, as well as problems with changing topologies. They propose to add a smoothed characteristic function in the integral layer architecture. This directly encodes the topology into the architecture, and it allows the architecture to continue to use FFT while allowing irregular grid inputs. The authors apply this domain agnostic method to the regular FNO as well as implicit FNO. The authors run experiments on material modeling, airfoil simulation, and material fractures.

**Strengths:**

- This is the first neural operator that can handle learning with topology changes. This is a useful property to handle fracture problems, such as material fracture or earthquakes.
- The technique is relatively simple: adding a characteristic function to the integral kernel and adding a smoothing term, but it is novel and elegant that the characteristic function encodes the topology directly in the architecture. This is how the authors’ technique can learn changing topologies, when, say Geo-FNO learns a fixed mapping and cannot learn changing topologies.
- The authors experiment on three problems: hyperelasticity, airfoils, and crack propagation, and they also perform ablation studies to separate the characteristic function and the smoothness.

**Weaknesses:**

- My biggest concern is that the technique may not be able to handle highly irregular topologies or topologies with fine-grained features, for two separate reasons. As shown in Figure 1, the resulting grid from the authors’ technique is still uniform, but with a characteristic added, to tell what is inside or outside the topology. Since the grid is still uniform, this may not work well for a topology that is very irregular or has fine features. This is in contrast to Geo-FNO, which learns a mapping, allowing it to handle highly irregular topologies and also to focus more grid points on the “important” parts of the topology and fewer grid points on the coarse / less important parts.
Furthermore, the smoothing function could magnify the above problem, making it even harder to model delicate parts of the topology. I think of this as a weakness because the authors say that handling irregular topologies is one of the main contributions of their work.
- The technique is fairly simple: it boils down to adding a characteristic function with smoothing. But on the other hand, this technique is the first to handle dynamically changing topologies.
- The authors did not release code or mention releasing code. If they authors released code (anonymously during submission) it would have a bigger impact.
- Why not compare airfoil with the other baselines? Also, why do the 3rd and 4th panels of Figures 4 look like there is a vertical line discontinuity in the center of the image?
- Why not compare the cracking to more baselines than just FNO, such as Geo-FNO?

**Questions:**

Can the authors comment on all of the weaknesses listed above?

Also, I wonder if Geo-FNO can be combined with DAFNO, to handle more complex topologies and also be able to handle changing topologies?

Do you plan to release your code? It could even be released during the rebuttal with e.g. https://anonymous.4open.science/.

I would raise my score if some or all of the points from the "weaknesses" section are addressed.

**Limitations:**

The authors discuss limitations in Section 5: they only consider changing domains where the grid is uniform, and they only consider problems with the same boundary conditions as the domain changes. I agree that it would be good to add a mapping for grid deformation in future work.

---

> ### Author Rebuttal · Authors · 2023-08-10
>
> We thank the reviewer for their valuable suggestions.
>
> **Highly irregular topologies or topologies with fine-grained features**: DAFNO can be readily combined with the grid mapping technique in Geo-FNO, to handle non-uniform grids. No modification on the NN architecture is required, and one just needs to include an analytical or trainable mapping from non-uniform/irregular mesh grids to uniform mesh grids. As a demonstration of this capability, we consider irregular grids in the airfoil problem and use a pre-computed function to map irregular grids to regular grids, then we train eDAFNO. In this irregular grid set, we place more grid points near the airfoil, so as to provide a better resolution near the important parts as suggested by the reviewer. The test error is provided in the table below: The test loss using the eDAFNO learned model is 0.659\%$\pm$0.007\%, which is similar to the DAFNO results on uniform grids. In Fig. 1 of the attached pdf file in the global response we demonstrate the irregular mesh, and in Figure 2 we plot the errors of Geo-FNO, DAFNO on regular grids, and DAFNO on irregular grids. One can see that while both DAFNOs substantially outperform Geo-FNO, the error contours from eDAFNO with irregular mesh show a smaller miss-match region near the airfoil, verifying the flexibility of DAFNO in meshing and its capability in resolving fine-grained features.
>
> **Smoothing function**: We treat the smoothing level $\beta$ as a hyperparameter, and it is tuned according to the validation error, to either smoothening the boundary or keep the original boundary untouched. This is illustrated in Fig. 8 in the appendix of the paper, where the original boundary is not smoothened with a very large smooth level $\beta$. As such, the $\beta$ as well as the smoothing level are automatically chosen based on the intrinsic resolution of the topology in training and validation datasets: if the topology prefers fine-grained features, a smaller $\beta$ (and smoother $\chi$) will result in a larger validation error and will not be chosen. We will include a discussion about this in the revised manuscript to strengthen the paper.
>
> **Release code**: We have uploaded our DAFNO code of the hyperelasticity problem on anonymous github and sent the link to the Area Chair. Our DAFNO package on other problems will also be made publicly available on github once the paper is accepted.
>
> **Airfoil with the other baselines**: The hyperelasticity problem serves as a thorough comparison against available baselines, after which we picked the baselines that have the best performance for the airfoil problem. This is in consistence with the settings in Geo-FNO. In order to further enrich the baselines, we have added another recent FNO variant, Factorized FNO (F-FNO), in both the elasticity and airfoil problems as a baseline. Additionally, as mentioned above, we added eDAFNO applied to irregular grids to highlight the fact that eDAFNO is not constrained to uniform grids. Our conclusion still holds. DAFNOs consistently outperform all selected baselines. The ``vertical line discontinuity'' in Fig. 4 is a physical phenomenon called shock wave, which reflects the nature of the transonic flow governed by the Euler's equation [1]. As discussed in [2] and [3], the capability of capturing such discontinuity is considered as an important metric in classical numerical methods.
>
> [1] Luk, Jonathan, and Jared Speck. "Shock formation in solutions to the 2D compressible Euler equations in the presence of non-zero vorticity." Inventiones mathematicae 214.1 (2018): 1-169.
>
> [2] Roe, Philip L. "Characteristic-based schemes for the Euler equations." Annual review of fluid mechanics 18.1 (1986): 337-365.
>
> [3] Hennemann, Sebastian, et al. "A provably entropy stable subcell shock capturing approach for high order split form DG for the compressible Euler equations." Journal of Computational Physics 426 (2021): 109935.
>
> **Cracking with Geo-FNO**: The crack example is presented intentionally as a problem where Geo-FNO falls short and cannot be used. The reason is that the Geo-FNO requires a pre-defined or trainable isomorphism between a rectangular domain and the targeted irregular domains. However, in fracture problems the domain undergoes severe topological changes including the emergence of new holes or discontinuities/boundaries (as shown in Fig. 5) which is not known a priori. Such an isomorphism does not exist (intuitively, there is no way to define a mapping between each yellow regions in Fig. 5 to a rectangular domain).
>
> Test errors for the hyperelasticity problem:
>
> | Model| 10 samples | 100 samples | 1000 samples |
> | :------------- | :-----------: | :-----------: | :-----------: |
> |eDAFNO | **16.446\%**$\pm$**0.472\%** | 4.247\%$\pm$0.066\% | **1.094\%**$\pm$**0.012\%**|
> |iDAFNO | 16.669\%$\pm$0.523\% | **4.214\%**$\pm$**0.058\%** | 1.207\%$\pm$0.006\%|
> |FNO w/ mask | 19.487\%$\pm$0.633\% | 7.852\%$\pm$0.130\% | 4.550\%$\pm$0.062\% |
> |IFNO w/ mask | 19.262\%$\pm$0.376\% | 7.700\%$\pm$0.062\% | 4.481\%$\pm$0.022\% |
> |Geo-FNO | 28.725\%$\pm$2.600\% | 10.343\%$\pm$4.446\% | 2.316\%$\pm$0.283\% |
> |GNO | 29.305\%$\pm$0.321\% | 18.574\%$\pm$0.584\% | 13.007\%$\pm$0.729\% |
> |DeepONet | 35.334\%$\pm$0.179\% | 25.455\%$\pm$0.245\% | 11.998\%$\pm$0.786\%|
> |F-FNO | 35.672\%$\pm$3.852\% | 12.135\%$\pm$5.813\% | 3.193\%$\pm$1.622\%|
> |UNet | 98.167\%$\pm$0.236\% | 34.467\%$\pm$2.858\% | 5.462\%$\pm$0.048\%|
>
> Results for the airfoil design problem:
> | Model | Train error | Test error |
> | :------------- | :-----------: | :-----------: |
> |eDAFNO | 0.329\%$\pm$0.020\% | **0.596\%**$\pm$**0.005\%** |
> |iDAFNO | 0.448\%$\pm$0.012\% | 0.642\%$\pm$0.020\% |
> |eDAFNO on irregular grids | 0.331\%$\pm$0.003\% | 0.659\%$\pm$0.007\% |
> |Geo-FNO | 1.565\%$\pm$0.180\% | 1.650\%$\pm$0.175\% |
> |F-FNO | 0.566\%$\pm$0.066\% | 0.794\%$\pm$0.025\% |
> |FNO w/ mask | 2.676\%$\pm$0.054\% | 3.725\%$\pm$0.108\% |
> |UNet w/ mask | 2.781\%$\pm$1.084\% | 4.957\%$\pm$0.059\% |

---

> > ### Comment · Reviewer_FUid · 2023-08-18
> >
> > Thank you for preparing the rebuttal to my review and the other reviews.
> >
> > I appreciate that you released code, added the F-FNO baseline, and ran airfoil with more baselines. I agree with the clarification of the shock wave phenomenon and about running baselines for airfoil and cracking. I also particularly like the example of airfoil with an irregular grid.
> >
> > Overall, I am impressed by the rebuttal and I raised my score.
> >
> > I have a few more questions. What is F-FNO? I think it is the same thing as tensorized FNO (TFNO)? In that case, I am wondering why the parameter count is higher for TFNO than for FNO? Did other hyperparameters change?
> >
> > I also wonder if DAFNO can be applied with the tensorized technique? And, although the new experiment with DAFNO on an irregular grid is great, it would be interesting to have DAFNO with a trainable mapping. It seems like a learnable irregular mapping would complement the domain adaptive part well? Note that I know it is only 4 days before the end of the discussion period, so this is not a request for more experiments, but a question, e.g., for future work.

---

> > > ### Author Response · Authors · 2023-08-19
> > > **Thank you!**
> > >
> > > We thank the reviewer for their kind response. We sincerely appreciate the reviewer's valuable time and suggestions, which helped us to improve the quality of this work.
> > >
> > > **Connections between F-FNO and T-FNO**: The reviewer is correct that F-FNO [1] has fundamentally similar architectures to TFNO [2] since they both adopt Fourier factorization, with some small differences such as the improved residual connection and training technique in F-FNO. The reason why F-FNO has more parameters than FNO (as also pointed out in the F-FNO paper) is that F-FNO used a bigger hidden size in the elasticity problem. To guarantee a fair comparison, we kept the original settings of F-FNO as in [1] since the total number of parameters are on a similar level as other methods.
> > >
> > > [1] Tran, Alasdair, et al. "Factorized Fourier Neural Operators." The Eleventh International Conference on Learning Representations. 2022.
> > >
> > > [2] Kossaifi, Jean, et al. "Multi-Grid Tensorized Fourier Neural Operator for High Resolution PDEs." (2022).
> > >
> > > **Future extensions on DAFNO**: We appreciate the reviewer's valuable suggestion. Both the tensorized architecture and the data-driven grid mapping should be readily applicable on DAFNOs. We agree with the reviewer that the combination of these two techniques with DAFNO would indeed be very interesting directions, which we will consider in the future work.

---

### Author Rebuttal · Authors · 2023-08-10

We thank the reviewers for the constructive comments, for recognizing the importance/usefulness of our work (reviewers FUid, UPhM, dnax), the novelty and elegance of DAFNO's architecture (reviewers FUid, fNa3, dnax, BzBv), DAFNO's role as the first neural operator that can handle dynamically changing topologies (reviewer FUid), effective architecture and strong experimental results (reviewers fNa3, dnax), computational advantage with far fewer parameters (reviewer BzBv), and clarity in writing (reviewers fNa3, dnax).

**Being simple yet flexible**: We want to comment on the simplicity of our DAFNO model: as also pointed out by Reviewer fNa3, being simple does not need to be seen as a negative point. Sometimes simpler is better! It is the performance that matters the most. The simplicity of DAFNO's architecture adds benefits in terms of facilitating implementation.

**Additional baselines and experiments**: In order to further enrich the baselines to compare to, we have added Factorized FNO (F-FNO) as another baseline in both the elasticity and airfoil problems. Our original conclusion still stands: DAFNOs consistently outperform all selected baselines. We also show the applicability of DAFNO on irregular/adaptive grids using the airfoil dataset, where the non-uniform grids are generated using sinusoidal functions and are highly adaptive in the vicinity of the airfoil. DAFNO achieves very similar accuracy compared to uniform meshes, while the error gets reduced near the airfoil.

**Release code**: We have uploaded our DAFNO code of the hyperelasticity problem on anonymous github and sent the link to the Area Chair. Our DAFNO package on other problems will also be made publicly available on github once the paper is accepted.

Test errors for the hyperelasticity problem, where bold numbers highlight the best method.

| Model| 10 samples | 100 samples | 1000 samples |
| :------------- | :-----------: | :-----------: | :-----------: |
|eDAFNO | **16.446\%**$\pm$**0.472\%** | 4.247\%$\pm$0.066\% | **1.094\%**$\pm$**0.012\%**|
|iDAFNO | 16.669\%$\pm$0.523\% | **4.214\%**$\pm$**0.058\%** | 1.207\%$\pm$0.006\%|
|FNO w/ mask | 19.487\%$\pm$0.633\% | 7.852\%$\pm$0.130\% | 4.550\%$\pm$0.062\% |
|IFNO w/ mask | 19.262\%$\pm$0.376\% | 7.700\%$\pm$0.062\% | 4.481\%$\pm$0.022\% |
|Geo-FNO | 28.725\%$\pm$2.600\% | 10.343\%$\pm$4.446\% | 2.316\%$\pm$0.283\% |
|GNO | 29.305\%$\pm$0.321\% | 18.574\%$\pm$0.584\% | 13.007\%$\pm$0.729\% |
|DeepONet | 35.334\%$\pm$0.179\% | 25.455\%$\pm$0.245\% | 11.998\%$\pm$0.786\%|
|F-FNO | 35.672\%$\pm$3.852\% | 12.135\%$\pm$5.813\% | 3.193\%$\pm$1.622\%|
|UNet | 98.167\%$\pm$0.236\% | 34.467\%$\pm$2.858\% | 5.462\%$\pm$0.048\%|
|FNO w/ smooth $\chi$ | 17.431\%$\pm$0.536\% | 5.479\%$\pm$0.186\% | 1.415\%$\pm$0.025\% |
|IFNO w/ smooth $\chi$ | 17.145\%$\pm$0.432\% | 5.088\%$\pm$0.146\% | 1.509\%$\pm$0.018\% |

Test errors for the airfoil problem.

| Model | Train error | Test error |
| :------------- | :-----------: | :-----------: |
|eDAFNO | 0.329\%$\pm$0.020\% | **0.596\%**$\pm$**0.005\%** |
|iDAFNO | 0.448\%$\pm$0.012\% | 0.642\%$\pm$0.020\% |
|eDAFNO on irregular grids | 0.331\%$\pm$0.003\% | 0.659\%$\pm$0.007\% |
|Geo-FNO | 1.565\%$\pm$0.180\% | 1.650\%$\pm$0.175\% |
|F-FNO | 0.566\%$\pm$0.066\% | 0.794\%$\pm$0.025\% |
|FNO w/ mask | 2.676\%$\pm$0.054\% | 3.725\%$\pm$0.108\% |
|UNet w/ mask | 2.781\%$\pm$1.084\% | 4.957\%$\pm$0.059\% |


The total number of parameters and per-epoch runtime (second) in the hyperelasticity problem.

|Model | eDAFNO | iDAFNO | FNO | IFNO | Geo-FNO | GNO | DeepONet | UNet | F-FNO |
| :------------- | :-----------: | :-----------: | :------------- | :-----------: | :-----------: | :------------- | :-----------: | :-----------: | :-----------: |
|nparams | 2.37M | 0.60M | 2.37M | 0.60M | 3.02M | 2.64M | 3.10M | 3.03M | 3.21M|
|runtime (s) | 2.00 | 1.70 | 1.81 | 1.62 | 5.12 | 98.37 | 940.12 | 5.04 | 3.41 |

---

### Decision · Program_Chairs · 2023-09-21

**Decision:**

Accept (poster)

**Comment:**

The paper presents an extension of the Fourier Neural Operator (FNO) designed to handle irregular grids and changing domain topologies while retaining the advantages of using the FFT, as in FNO. To achieve this, the authors adapt the FNO kernel by introducing a smoothed characteristic function that encodes domain transformations. This enables the method to generalize to new geometries. The approach is evaluated in the contexts of material modeling, airfoil simulation, and a dataset of material fracture evolution.


The extension of FNO to accommodate irregular grids has inspired several recent extensions. Reviewers agree that the approach proposed here is novel, simple, and efficient. In response to the reviewers' suggestions, the authors have conducted additional experiments that further support their findings. All the reviewers believe that this represents a valuable enhancement for neural operator deployment and recommend acceptance.